# Tissue and liquid biopsy profiling reveal convergent tumor evolution and therapy evasion in breast cancer

Smruthy Sivakumar [1] ✉, Dexter X. Jin[1], Hanna Tukachinsky[1], Karthikeyan Murugesan [1], Kimberly McGregor[1], Natalie Danziger [1], Dean Pavlick[1], Ole Gjoerup[1], Jeffrey S. Ross[1,2], Robert Harmon[1], Jon Chung[1], Brennan Decker[1], Lucas Dennis[1], Garrett M. Frampton [1], Luciana Molinero[3], Steffi Oesterreich [4], Jeffrey M. Venstrom[1], Geoffrey R. Oxnard[1], Priti S. Hegde[1] & Ethan S. Sokol [1] ✉

Pathological and genomic profiling have transformed breast cancer care by matching patients to targeted treatments. However, tumors evolve and evade therapeutic interventions often through the acquisition of genomic mutations. Here we examine patients profiled with tissue (TBx) and liquid biopsy (LBx) as part of routine clinical care, to characterize the tumor evolutionary landscape and identify potential vulnerabilities in the relapsed setting. Real-world evidence demonstrates that LBx is utilized later in care and identifies associations with intervening therapy. While driver events are frequently shared, acquired LBx alterations are detected in a majority of patients, with the highest frequency in ER+ disease and in patients with longer biopsy intervals. Acquired mutations are often polyclonal and present at lower allelic fractions, suggesting multi-clonal convergent evolution. In addition to well-characterized resistance mutations (e.g., *ESR1, NF1, RB1, ERBB2*), we observe a diversity of rarer but potentially targetable mutations (e.g., *PIK3CA, HRAS/NRAS/KRAS, FGFR1/2/3, BRAF*) and fusions (e.g., *FGFR1/2, ERBB2, RET*), as well as *BRCA1/2* reversions through a variety of mechanisms, including splice alterations and structural deletions. This study provides insights on treatment and selection-driven tumor evolution and identifies potential combinatorial treatment options in advanced breast cancer.

A growing armamentarium of molecular diagnostics are offering an opportunity to enable precision therapies for patients with advanced breast cancer. While testing for hormone receptor status and HER2 amplification have long been standard, next-generation sequencing-based diagnostics are now available to guide the use of precision therapies, including PI3K inhibition[1] and PARP inhibition[2].

Furthermore, recent advances for patients with breast cancer, including antibody-drug conjugates[3,4] as well as immune checkpoint inhibitors[5,6] have required a reconsideration of the optimal diagnostic approach for advanced breast cancer. While these therapies have demonstrated promise in terms of breadth and durability of response with some therapy options moving from metastatic to early disease

[1]Foundation Medicine, Inc., Cambridge, MA, USA. [2]SUNY Upstate Medical University, Syracuse, NY, USA. [3]Genentech, Inc., South San Francisco, CA, USA. [4]Department of Pharmacology and Chemical Biology, University of Pittsburgh, Pittsburgh, PA, USA. ✉e-mail: ssivakumar@foundationmedicine.com; esokol@foundationmedicine.com

settings[3], cancer continues to evolve and evade therapeutic pressure over time[7–14].

on-pathway or bypass mechanisms. For endocrine therapy resistance, for instance, Razavi and colleagues identified a 18% prevalence of on-target resistance via the acquisition of *ESR1* point mutations in addition to recurrent alterations in bypass pathways such as MAPK and MYC, although the diversity of resistance mechanisms has yet to be fully described[9]. Clinically, resistance alterations have the potential to inform treatment decisions. For example, the acquisition of *ESR1* point mutations may motivate use of existing or emerging selective estrogen receptor degraders (SERDs) and acquired *PIK3CA* alterations are potentially targetable with PI3K inhibitors.

Multifocal sequencing has revealed significant heterogeneity in breast cancer during development, evolution, and therapy resistance[15–17]. Different clonal populations may respond differently to therapy and independently acquire cancer-driving and resistance alterations. Zundelevich and colleagues identified significant variability in the genomic alterations in primary, multiple relapse, and metastatic biopsies in patients resistant to endocrine therapy[17]. Such heterogeneity poses a challenge clinically for ideal treatment of patients with multifocal disease and diagnostically to identify the relevant drivers and resistance mutations.

Liquid biopsies (LBx), which interrogate circulating tumor DNA (ctDNA) in the blood, have emerged as a diagnostic and monitoring tool in breast and other cancer types. This platform provides genomic information on dozens to hundreds of genes implicated in cancer, with some next-generation sequencing (NGS)-based LBx now having received regulatory approval both inside (FoundationOne® Liquid CDx, Guardant360® CDx) and outside the US (FoundationOne® Liquid CDx). LBx has particular utility in the metastatic setting where high shed of ctDNA in the plasma may represent an option when tissue biopsy (TBx) material may not be available or representative of multifocal disease states in which metastatic subclones may acquire distinct resistance mutations[18,19].

In this study, we examined breast tumors profiled with TBx and LBx comprehensive genomic profiling (CGP) as part of routine clinical care to characterize the diversity of acquired mutations and their associations with therapeutic interventions in a real-world setting (Fig. 1).

## Results

### Mutational spectrum of breast tumors profiled with tissue and liquid biopsy

We examined the landscape of genomic mutations identified in a large, real-world database of breast cancers profiled with TBx ($n = 29,704$) and LBx ($n = 3339$) CGP (Fig. 2, Supplementary Data 1, Fig. 3a, b). The genes commonly targeted in both the assays are presented in Supplementary Data 2. LBx and TBx identified largely similar prevalence for short variants (point mutations and short indels) in most genes ($p > 0.05$; Supplementary Data 3; Fig. 3a, b). Both platforms identified an appreciable frequency of activating alterations in *PIK3CA* mapping to alpelisib companion diagnostic variants[1] (27.5% in TBx and LBx; Fig. 3c). Further, 4.5% of TBx and 7.2% of LBx samples harbored 2 or more *PIK3CA* mutations, possibly sensitizing to PI3K inhibitors, as described previously[20]. Deleterious *BRCA1/2* alterations were also observed at similar frequencies across the two platforms (7.1% TBx, 7.4% LBx; Fig. 3d). Notably, there was a higher prevalence in LBx for *ESR1* (10.6% TBx, 25.9% LBx), *NF1* (4.5%, 8.1%), *KRAS* (2.0%, 4.1%), *RB1* (4.6%, 8.0%), and *ERBB2* mutations (3.4%, 4.3%), alterations known to be associated with therapeutic resistance[9–12,14,21] (Supplementary Data 3; Fig. 3a, b). A comparison of point mutations and indels detected across different receptor subgroups between the two assay platforms showed largely similar patterns of gene alterations; statistically significant differences included elevated prevalence of *ESR1* and *FGFR1* in ER+ LBx, and *PTEN* in TNBC LBx (Supplementary Fig. 1, Supplementary Fig. 2, Supplementary Data 4). Overall, LBx showed a lower sensitivity for copy number alterations, with differences in *MYC*, *FGFR2*, and *ERBB2* amplifications (Supplementary Fig. 3; Supplementary Data 5). Both platforms also identified similar patterns of co-occurring gene alterations, such as the co-occurrence of *TP53* with *BRCA1* and *PIK3CA* with *ESR1* (Supplementary Data 6; Supplementary Fig. 4). In an analysis of the most recent assay targeting 324 genes ($n = 1430$)[22], patterns of gene alteration prevalence were similar to

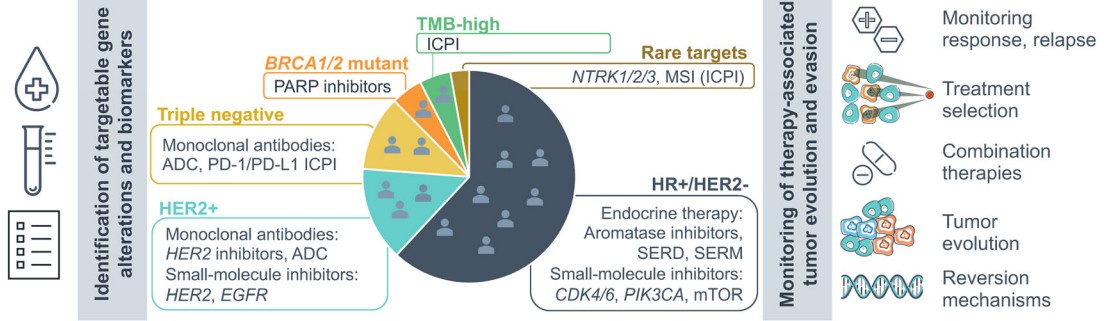

SERD: Selective Estrogen Receptor Degrader
SERM: Selective Estrogen Receptor Modulator
TMB: Tumor mutational burden
MSI: Microsatellite instability
ICPI: Immune checkpoint inhibitor
ADC: Antibody drug conjugate

**Fig. 1 | Graphical outline of targeted therapies in breast cancer and the clinical utility of liquid biopsy profiling during the course of care.** Liquid biopsies may be utilized in identifying targetable gene alterations and biomarkers for specific therapies across different subtypes of breast cancer as well as in monitoring therapy-associated implications on tumor evolution and treatment evasion. (SERD Selective Estrogen Receptor Degrader, SERM Selective Estrogen Receptor Modulator, TMB tumor mutational burden, MSI microsatellite instability, ICPI immune checkpoint inhibitor, ADC antibody-drug conjugate).

## Comprehensive genomic profiling (CGP) breast cancer cohort

| | TBx | LBx | Paired |
|---|---|---|---|
| **Number of patients/samples** | 29,704 | 3,339 | 712 |
| **Age median [IQR], years** | 58 [48 - 66] | 64 [55 - 71] | 56 [47 - 64] |
| **TMB median [IQR], mut/mb** | 3.5 [1.7 - 5.2] | | 2.6 [1.3 - 5.2] |
| **LBx tumor fraction median [IQR], %** | | 2.6 [0.5 - 18.5] | 2.7 [0.4 -18.5] |
| **Tissue tumor purity median [IQR], %** | 50 [34 - 67] | | 50 [31 - 66] |
| **Time between biopsies range (median), days** | | | 0 – 6,455 (495) |

| | | Prevalence count (%) | LBx tumor fraction median % |
|---|---|---|---|
| **Histology** | IDC | 276 (83.9) | 1.8 |
| | ILC | 45 (13.7) | 0.9 |
| | Metaplastic | 8 (2.4) | 12.2 |
| | Other/NOS | 383 | 3.6 |
| **Site** | Local | 252 (35.4) | 1.1 |
| | Metastatic | 310 (43.5) | 4.0 |
| | Lymph node | 87 (12.2) | 2.6 |
| | Unknown | 63 (8.9) | 2.6 |
| **Subtype** | ER+/HER2- | 301 (63.6) | 2.8 |
| | ER-/HER2+ | 33 (7.0) | 1.3 |
| | ER+/HER2+ | 37 (7.8) | 2.1 |
| | TNBC | 102 (21.6) | 5.7 |
| | Other/Unavailable | 239 | 2.1 |

## Clinico-genomic database (CGDB) breast cancer cohort

| | | | TBx | LBx |
|---|---|---|---|---|
| | **Total patients \| Total specimens** | | 6,757* \| 7,110 | 1,150* \| 1,228 |
| **PATIENT LEVEL** | **Sex** | Female | 6,679 (98.9) | 1,144 (99.5) |
| | | Male | 77 (1.1) | 6 (0.5) |
| | | *Unknown* | 1 | - |
| | **Race** | Asian | 126 (2.0) | 31 (3.0) |
| | | African American/Black | 638 (10.3) | 85 (8.1) |
| | | Hispanic/Latino | 15 (0.2) | 0 (0.0) |
| | | White | 4,405 (70.9) | 726 (69.5) |
| | | Other | 1,028 (16.6) | 202 (19.4) |
| | | *Unknown* | 545 | 106 |
| | **Practice type** | Academic | 645 (9.5) | 170 (14.8) |
| | | Community | 6,112 (90.5) | 980 (85.2) |
| | **Histology** | Inflammatory | 47 (1.1) | 7 (1.0) |
| | | IDC | 3,662 (83.0) | 599 (81.5) |
| | | ILC | 504 (11.4) | 102 (13.9) |
| | | Medullary | 3 (0.0) | 0 (0.0) |
| | | Metaplastic | 60 (1.4) | 4 (0.5) |
| | | Mixed | 105 (2.4) | 18 (2.5) |
| | | Mucinous | 19 (0.4) | 4 (0.5) |
| | | Papillary | 11 (0.3) | 1 (0.1) |
| | | Tubular | 2 (0.0) | 0 (0.0) |
| | | *Unknown* | 2,344 | 415 |
| **SPECIMEN LEVEL** | **Median age at reported date [IQR]** | | 60.0 [50.0-68.0] | 63.0 [55.0-71.0] |
| | **ECOG performance status before reported date** | 0 | 1,437 (40.1) | 231 (37.5) |
| | | 1 | 1,607 (44.9) | 283 (45.9) |
| | | 2+ | 535 (15.0) | 102 (16.6) |
| | | *Unknown* | 3531 | 612 |
| | **Number of lines of therapy before reported date** | 0 | 1,259 (19.5) | 163 (14.7) |
| | | 1 | 1,616 (25.1) | 220 (19.8) |
| | | 2 | 1,074 (16.7) | 204 (18.3) |
| | | 3+ | 2,491 (38.7) | 525 (47.2) |
| | | *Unknown* | 670 | 116 |
| | **Subtype status (ER, PR, HER2)** | ER+/HER2- | 3,826 (62.1) | 766 (73.9) |
| | | ER-/HER2+ | 304 (4.9) | 36 (3.4) |
| | | ER+/HER2+ | 858 (13.9) | 144 (13.9) |
| | | TNBC | 1,175 (19.1) | 91 (8.8) |
| | | Other/Unavailable | 947 | 191 |
| | **Physician-reported stage at time of ordering of FMI CGP test** | 0 | 12 (0.4) | 3 (0.4) |
| | | I | 119 (4.1) | 25 (3.5) |
| | | II | 298 (10.3) | 37 (5.1) |
| | | III | 317 (10.9) | 31 (4.3) |
| | | IV | 2,152 (74.3) | 628 (86.7) |
| | | Unknown | 4,212 | 504 |

\* Some patients received both TBx and LBx testing

**Fig. 2 | Overview of the study cohort and patient characteristics.** This study cohort comprised patients who received comprehensive genomic profiling (CGP) as part of routine clinical care (top). A total of 29,704 patients with tissue biopsy (TBx)-based CGP, 3339 patients with liquid biopsy (LBx)-based CGP and 712 patients with longitudinal TBx and LBx CGP were examined. Patient and specimen-level characteristics have been provided for each of these cohorts. Additional breakdown of histological subtype, site of biopsy and receptor subtype information is provided for the 712 patients with longitudinal TBx and LBx CGP. Treatment information was retrospectively reviewed in an independent de-identified clinico-genomic database (CGDB), comprising 6,757 patients with TBx CGP and 1150 patients with LBx CGP (bottom). Patient and specimen-level characteristics have been provided for each of these cohorts. Additional information such as ECOG status, lines of therapy are available for samples in the CGDB cohort. (ILC invasive lobular carcinoma, IDC invasive ductal carcinoma, TNBC triple-negative breast cancer, NOS not otherwise specified, IQR interquartile range).

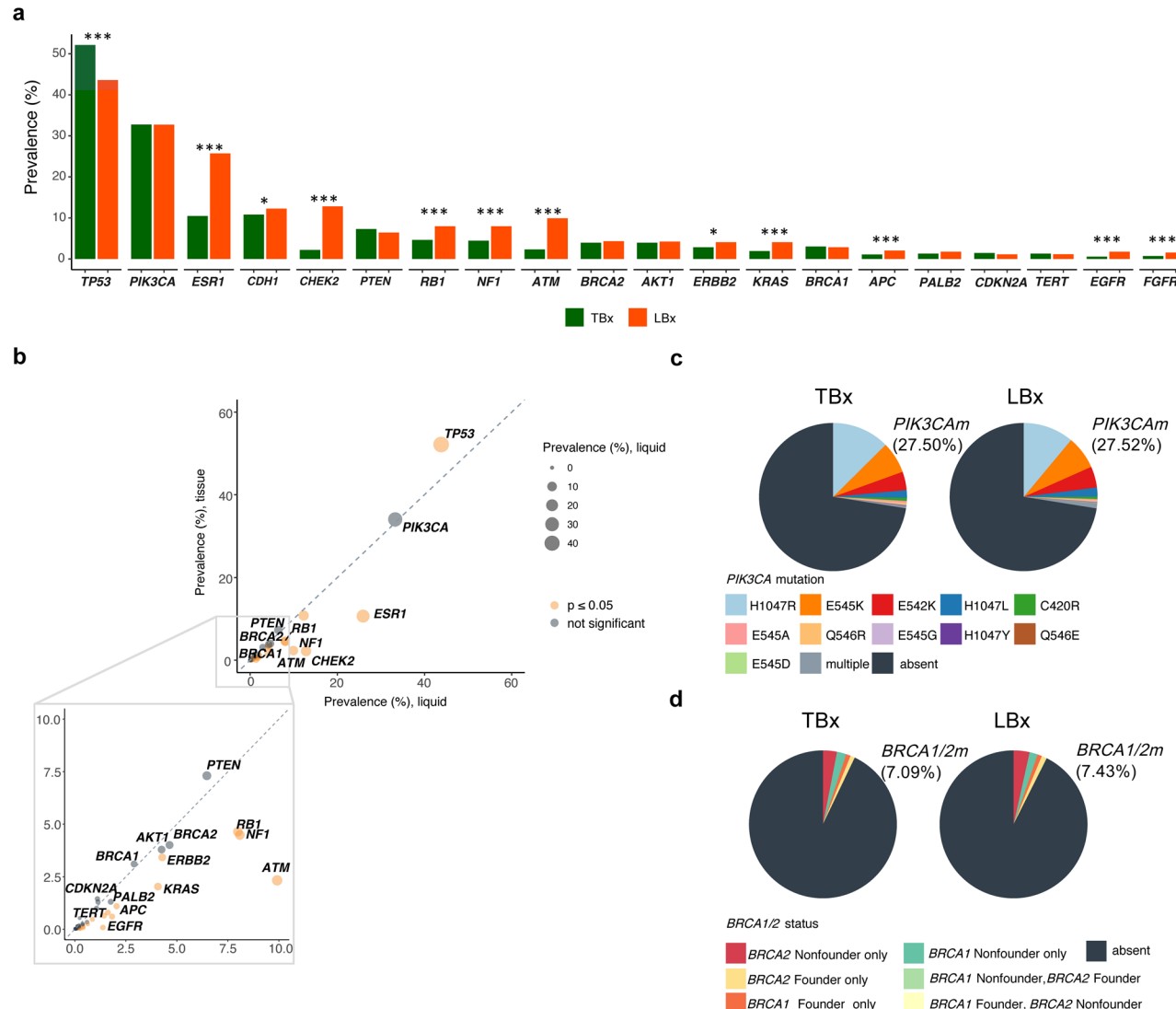

**Fig. 3 | Mutation landscape from comprehensive genomic profiling in 29,704 tissue biopsy (TBx) and 3339 liquid biopsy (LBx) samples in breast cancer. a** Prevalence of the 20 most frequently altered genes. Alteration prevalence between the two assays was compared using a two-sided Fisher's exact test with FDR correction (*p*-value thresholds *: 0.05, **: 0.01, ***: 0.001). **b** Comparison of prevalence for all genes assessed in TBx (*y*-axis) and LBx (*x*-axis), with a zoomed in view of alterations seen at frequency <10%. Genes with statistically significant alteration prevalence between the two assays, based on a two-sided Fisher's exact test with FDR correction (*p*-value ≤ 0.05) have been shown in orange. Pie chart showing the breakdown and prevalence of different classes of. **c** *PIK3CA* and **d** *BRCA1/2* alterations across the two biopsy platforms. Only short variants (e.g., point mutations and short indels) were analyzed here.

those observed in the larger LBx cohort (*n* = 3339), particularly for samples with tumor fraction of ≥10%. (Supplementary Fig. 5, Supplementary Data 7).

Examination of 6757 patients profiled with TBx and 1150 patients profiled with LBx in a clinico-genomic database (CGDB) revealed differences in the ordering patterns for these CGP tests (Fig. 2). Compared to TBx, patients profiled with LBx were older (63 y vs. 60 y, *p* < 10⁻⁵), more likely to have metastatic disease (86.7% stage IV vs. 74.3% for TBx, *p* < 10⁻⁵), and were treated with more lines of prior therapy (median 3+ vs. 2, *p* < 10⁻⁵), suggesting that LBx is frequently used in more advanced setting. Consistent with these differences, LBx showed higher similarity with metastatic-biopsied TBx relative to breast-biopsied TBx in the overall cohort, especially for genes like *ESR1, NF1*, and *RB1* (Supplementary Fig. 6; Supplementary Data 3). This suggests that some of these differences in alteration prevalence may be attributed to disease stage. However, since most resistance-associated alterations were still observed at a significantly higher prevalence in LBx compared to metastatic-biopsied TBx, other factors

are likely influencing these findings, including the number of lines of prior therapy.

## 712 patients with longitudinal TBx and LBx profiling reveal a spectrum of acquired tumor resistance mechanisms and actionable targets

712 patients were profiled with LBx following a TBx, during the course of care (Fig. 2, Supplementary Data 8). Samples were temporally heterogeneous, with a median of 495 days between biopsy collection (range 0–6455 days), included a mix of local (35%) and metastatic TBx (44%), and represented a diversity of subtypes (histological: IDC 84%, ILC 14%, metaplastic 2%; pathological: ER+/HER2− 64%, HER2+15%, TNBC; 22%) (Fig. 2). For pathogenic mutations detected in TBx, positive percent agreement (PPA) was 72% for LBx, heavily influenced by estimated tumor fraction and predicted germline status (Supplementary Fig. 7, Supplementary Data 9). PPA was highest in samples with a tumor fraction of >10%, especially for potentially actionable alterations in *PIK3CA* (93%) and *BRCA1/2* (95%) (Supplementary Fig. 7a–c,

Supplementary Data 9). PPA was higher for predicted germline alterations relative to somatic alterations (100% vs 70%; Supplementary Fig. 7d, Supplementary Data 9), such as in cases with founder *BRCA1/2* alterations (Supplementary Fig. 7e). Of note, germline alterations were detected, regardless of tumor fraction, as expected since the former can be detected in non-tumor DNA even when ctDNA shed is low (Supplementary Fig. 7d, Supplementary Fig. 8). Tissue biopsies were further examined based on the site of TBx: local (breast biopsy), metastatic (distant site biopsy) and lymph node. Due to increased ctDNA shedding, PPA was significantly higher in metastatic and lymph node biopsies compared to breast-biopsied tumors (80% vs. 78% vs. 58%, respectively, $p < 10^{-4}$, Supplementary Fig. 7f). PPA was not impacted by receptor subtype or time between tests (Supplementary Fig. 7g, Supplementary Fig. 7h, Supplementary Data 9).

We examined the landscape of acquired mutations in follow-up LBx and identified events most commonly in *TP53* (246 alterations; 26.8% of acquired alterations), *ESR1* (181; 19.7%), *NF1* (69; 7.5%), *PIK3CA* (65; 7.1%), *BRCA2* (48; 5.2%), *PTEN* (39; 4.2%), *RB1* (32; 3.5%), and *ERBB2* (21; 2.3%) (Fig. 4a). However, there was also a long tail of acquired mutations with acquired short variants seen in 48 genes, including in *KRAS* ($n = 13$), *AKT1* ($n = 9$), *EGFR* ($n = 7$), *FGFR2* ($n = 7$), *BRAF* ($n = 5$), *MTOR* ($n = 4$), *FGFR1* ($n = 4$), *NRAS* ($n = 3$), *HRAS* ($n = 2$), *FGFR3* ($n = 1$), and *RET* ($n = 1$) (Supplementary Data 10). Acquired short variants were more prevalent in patients with a longer time between biopsies, with LBx collected 3+ years after TBx harboring at least one acquired alteration = 69.5% of cases (Fig. 4b, Supplementary Data 11). A longer duration between biopsies likely represents a larger number of intervening therapies, which may result in the development of resistance mutations and more dynamic tumor evolution.

When stratified based on molecular subtype, acquired alterations were most common in ER+/HER2− samples (60%) with lower frequencies in TNBC (51%) and HER2+ subgroups (51% in ER+/HER2+ , 33% in ER−/HER2+ ) (Fig. 4c, Supplementary Fig. 9, Supplementary Data 8, Supplementary Data 11). These findings are consistent with previous reports identifying high frequencies of resistance alterations in patients with ER+ subtype treated with endocrine therapy[8,9,23].

Although less common, liquid biopsies also detected acquired amplifications and rearrangements ($n = 77$). These included recurrent amplifications (Fig. 4d) in *FGFR1* ($n = 9$), *CD274* (PD-L1; $n = 5$), *PDCD1LG2* (PD-L2; $n = 4$), *MYC* ($n = 3$), *RET* ($n = 3$), *EGFR* ($n = 2$), *ERBB2* ($n = 2$), *ESR1* ($n = 2$), *KRAS* ($n = 2$), and *PIK3CA* ($n = 2$) and rearrangements (Fig. 4e) involving *NF1* ($n = 5$), *BRCA1* ($n = 4$), *BRCA2* ($n = 4$), *TP53* ($n = 4$), *RB1* ($n = 3$), *CDK12* ($n = 2$), *FGFR2* ($n = 2$), and *PTEN* ($n = 2$) (Supplementary Data 10). Of note, liquid biopsy samples with acquired amplifications and rearrangements had a significantly higher tumor fraction compared to samples where these large, acquired events were not detected (28.6%, 26.5% vs. 2.0% respectively, each $p < 10^{-5}$; Supplementary Fig. 10), thereby highlighting the potential of high tumor fraction LBx samples in detecting these complex events. When including all classes of mutations, acquired alterations were seen in 56% of cases, including 70% of ER+/HER2− cases (Supplementary Data 8).

Many of the acquired alterations represent potentially targetable entities. Although acquired *PIK3CA* mutations were less often found in canonical codons 542/545/1047 (35/65, 54% acquired vs 134/184, 73% shared; $p = 0.008$) and included E726K (9/65), E418K (3/65), and E53K (3/65) mutations, most acquired alterations were observed in contexts qualified for PI3K inhibitors (54%) (Supplementary Data 12). For the ER+/HER2− population, acquired *PIK3CA* alterations were observed in 8% of patients. Previous reports have associated acquired *PIK3CA* alterations with APOBEC mutagenic processes[18]; consistent with this, 77% (48/62) of acquired missense *PIK3CA* alterations were observed in canonical APOBEC trinucleotide contexts (T[C > T/G]N) (Supplementary Data 12). Acquired *ERBB2* short variants, which are potentially targetable with HER2 TKIs or with HER2 antibody-drug conjugates,

were common in the dataset ($n = 21$ alterations in 15 patients) and were observed in both the ER+/HER2− and HER2+ subsets (2% and 4%, respectively)[24–27]. *ERBB2* mutations have been previously reported as an endocrine therapy resistance mechanism, but their presence in the HER2+ population may represent resistance to HER2-targeted therapies[9,28]. In addition to short variants, a patient with ER−/HER2+ breast cancer also exhibited an acquired *ERBB2-GRB7* fusion in the follow-up LBx, which retained the extracellular, juxtamembrane, and kinase domains.

Follow-up LBx also detected rare, potentially targetable acquired alterations. Alterations in *FGFR1-3* family members were observed in 3% (23/712) of cases, predominantly in ER+ disease (Supplementary Fig. 11), and included amplifications of *FGFR1* ($n = 9$ cases) and *FGFR1/2* fusions ($n = 3$ cases). *KRAS* alterations were acquired in 14 cases, including nine with an acquired alteration in codon 12. A *RET-BAIAP2L1* fusion was identified in LBx of patient 411, 1591 days after TBx (Fig. 4e, Supplementary Data 8, Supplementary Data 10); previous studies have demonstrated a role of *RET* in endocrine resistance with possible responses to *RET* inhibitors[29–31].

follow-up LBx, collected 132 days after the TBx, with a truncal *BRCA2* A938fs\*21 mutation (Fig. 4f). Similarly, most patients had multiple subclonal *BRCA1/2* reversion events, likely representing convergent evolution in the context of acquired tumor resistance through multiple subclones. There were similar reversion rates in *BRCA1* and *BRCA2* (25% v 19%, $p = 0.7$). *BRCA2* reversions often comprised a larger number of events (median = 3.5 reversion events) compared to *BRCA1* (median = 1 reversion event). Notably, five of the 10 cases with a predicted *BRCA1/2* reversion mutation also exhibited a large structural rearrangement event (Fig. 4f), highlighting the importance of broad LBx profiling in the resistance setting. Of note, we also examined patients with ≥2 *BRCA1* or *BRCA2* mutations in the overall LBx cohort ($n = 3339$), and identified 8 out of 10 cases with ≥2 *BRCA1*, and 16 out of 23 cases with ≥2 *BRCA2* showing mechanisms of reversions (Supplementary Data 13).

Consistent with tumor heterogeneity and clonal evolution dynamics, acquired short variants in LBx revealed a lower clonal fraction relative to shared alterations with TBx (Fig. 4g). These findings highlight the utility of LBx in identifying the repertoire of subclonal events, present in only a portion of the tumor or a subset of metastatic lesions. In addition, acquired alterations tend to be polyclonal (Fig. 4h), across different receptor subtypes (Supplementary Fig. 12a), with over 50% of *PIK3CA* and *TP53*-mutant samples exhibiting 2 or more alterations. Similarly, polyclonal alterations in *BRCA1, ESR1, ERBB2,* and *BRCA2* were also commonly identified. In contrast, alterations in *KRAS* and *CDH1* often comprised single events. Presence of polyclonal mutations was also frequent in the overall LBx cohort, especially in *ESR1, TP53, PIK3CA,* and *NF1* (~20%+), whereas, other gene alterations, such as in *AKT1, KRAS,* and *EGFR* were commonly monoclonal (Supplementary Fig. 12b). These polyclonal alterations allude to mechanisms of multiclonal convergent evolution in acquiring resistance. To test for sub-threshold evidence in the matched TBx, we examined 381 variants detected uniquely in the LBx; 363 (>95%) had ≤ five supporting reads, within the margin of sequencing error (Supplementary Data 14). These findings further support that a majority of these alterations are truly acquired, likely resistance mutations detected in the follow-up LBx sample.

To confirm the patterns observed in the matched TBx/LBx setting, we independently assessed 1328 patients with breast cancer who received longitudinal TBx CGP. A similar landscape of acquired alterations (*TP53, ESR1, NF1, BRCA1/2, ERBB2*), associations with time between biopsies, and elevated prevalence of acquired alterations in the ER+/HER2− subtype were identified in the second TBx (Supplementary Fig. 13, Supplementary Data 15, Supplementary Data 16). Interestingly, a higher rate of acquired events in multiple genes was observed in follow-up LBx compared

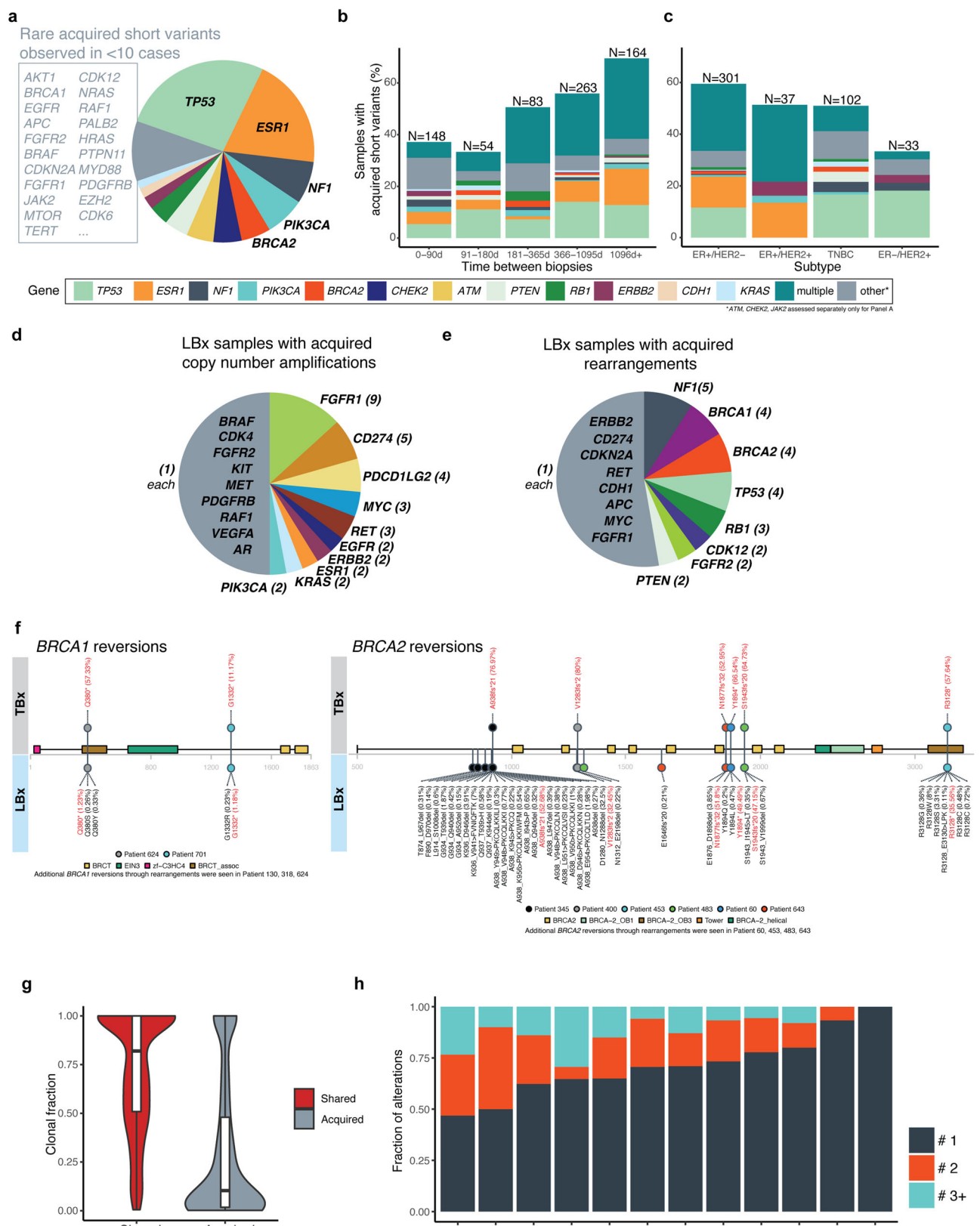

to TBx (150 follow-up LBx samples, 21% vs. 98 follow-up TBx samples, 7.3%; $p < 10^{-5}$, Supplementary Data 11, Supplementary Data 16). These results suggest that sequential biopsies, whether from liquid or tissue, can better inform mechanisms of tumor evolution and potential resistance.

Clonal hematopoiesis (CH), an age-related process of accumulating somatic mutations in hematopoietic stem cells, is a known layer of complexity in LBx-based profiling. Alterations frequently associated with CH, such as in *JAK2* (specifically, V617F; $n = 4$ cases), *ATM* ($n = 44$ alterations in 34 cases), and *CHEK2* ($n = 46$ alterations in 36 cases;

**Fig. 4 | Liquid biopsies identify acquired alterations associated with therapeutic resistance. a** Acquired short variants (e.g., point mutations, short indels) in patients with longitudinal tissue biopsy (TBx)/liquid biopsy (LBx), where LBx was collected at the time of or following TBx (*n* = 712). The fraction of samples with acquired short variants based on **b** collection time difference and **c** receptor subtype. **d** Acquired copy number amplifications detected in follow-up LBx samples (number of detected events for each gene is provided within parenthesis). **e** Acquired rearrangements detected in follow-up LBx samples (number of detected events for each gene is provided within parenthesis). **f** Reversion mechanisms impacting *BRCA1* and *BRCA2*. Each patient is represented by a unique color, with shared *BRCA1/2* alterations shown in red and identified reversions shown in black. Variant allele frequency for each identified *BRCA1/2* short variant is provided. **g** Distribution of the estimated clonal fraction for shared and acquired short variants in LBx. Each box plot displays the interquartile range (IQR), with the lower and upper boundaries representing the 25th and 75th percentile; the line within the box represents the median and the whiskers extend to ±1.5 × IQR. **h** Polyclonality of acquired short variants identified in LBx. The fraction of a single (#1) or multiple (#2, #3+) detected alterations within each gene is shown.

Supplementary Data 9), were also observed to be unique to LBx in our cohort, albeit rare. These alterations were observed at a higher frequency in LBx relative to TBx (1.4% v 0.1% *JAK2*, 9.9% v 2.3% *ATM*, 12.8% v 2.2% *CHEK2* each *p* < 0.05; Supplementary Data 2), at lower allelic fractions (median 0.4% v 1.2% for all mutations, *p* < 0.05), and trended older (63 y vs 58 y overall), consistent with CH origin[34]. Though *TP53* could also be associated with CH, small differences were seen with patient age (median 59 y v 58 y) and allelic fraction (median 0.9% v 1.2% overall; *p* > 0.05). Furthermore, *TP53* prevalence was *lower* in LBx relative to TBx (43.8% v 52.2%, *p* < 10$^{-5}$; Supplementary Data 2). *NF1* has also been reported as a CH-associated gene[35]. While *NF1* alterations were higher in prevalence in LBx compared to TBx (8.1% v 4.5%), *NF1* is also a known resistance mechanism to endocrine therapy[9,10,36]. Consistent with this, *NF1* was frequently acquired in follow-up tissue biopsies in the overall cohort (Supplementary Fig. 13), and was observed frequently in samples taken post-endocrine therapy in our clinical cohort, as will be described below. Notwithstanding these rare, potential CH-associated mechanisms, our findings showcase the diversity of the genomic landscape derived from LBx, thereby providing insights to better understand the tumor evolution dynamics and guide therapy selection by offering promising targets for patients with advanced breast cancers.

## Acquired alterations are associated with therapeutic intervention

We next examined the association of acquired alterations with treatment patterns using a clinico-genomic database (CGDB) comprising 196 patients with longitudinal genomic tests, eligible for analysis (Fig. 5a). Similar to the CGP cohort, acquired alterations were most commonly observed in *TP53* (30.1%), *ESR1* (20.3%), *PTEN* (15.4%), *NF1* (12.6%), *PIK3CA* (10.5%), *ATM* (3.5%), *BRCA1* (4.2%), *RB1* (3.5%), *KRAS* (2.8%), *ERBB2* (2.1%), and *BRCA2* (2.1%) (Fig. 5b, Supplementary Data 17).

co-treatment with endocrine therapy, also presented with acquired *ESR1*, *NF1*, and *PIK3CA* alterations. HER2-targeted therapies were associated with acquired alterations in *ERBB2, FGFR2, KRAS*, and *PIK3CA*. These findings further elucidate the treatment-specific spectrum of acquired alterations and clonal evolution, inferred through longitudinal profiling. When leveraging the overall clinical cohort to examine the prevalence of gene alterations in patient exposed or not exposed to a given therapy class, similar associations were observed, such as elevated levels of *ESR1* and *RB1* in patients receiving a combination of endocrine therapy and CDK4/6 inhibitor therapy, and *BRCA1/2* reversions following PARPi therapy (Supplementary Fig. 14, Supplementary Data 18). A number of additional, rare gene alterations were elevated in post-treatment samples of specific therapy classes (Supplementary Data 18, Supplementary Fig. 14).

We more closely examined the treatment journeys of three patients with multiple acquired alterations (Fig. 5c).

### Case 1
A 67-year-old woman presented with Stage IV ER+/PR+/HER2− IDC. Tissue CGP from a breast mass identified *IGF1R* amplification and a *GATA3* frameshift. Following progression on letrozole/palbociclib the patient was switched to fulvestrant/abemaciclib without progression. Liquid CGP during fulvestrant/abemaciclib treatment detected CDK4/6 inhibitor resistance mutation in *RB1* (variant allele fraction, VAF 3.8%) and endocrine therapy resistance alteration *ESR1* Y537S (0.4%). Other variants detected included *ATM* (VAF 12%), *CDK12* (1.4%), *TP53* (0.2%), and *EGFR* A1118T (0.2%). Mutations in *ATM* are common in solid tumors but are also in the bone marrow niche, raising the possibility of CH origin, although it is uncommon to detect CH variants above 10% of the cell-free DNA content[37].

### Case 2
A 41-year-old woman presented with Stage IIB ER+/HER2− ILC and progressed with skin metastases at age 51 following bilateral mastectomy and adjuvant tamoxifen/letrozole. Tissue CGP revealed a L1620*fs in exon 10 of *BRCA1* (predicted somatic, with loss of heterozygosity of the wild type allele), and amplifications of *MDM2* and *FRS2*. The patient progressed on fulvestrant/palbociclib, anastrozole/abemaciclib, and talazoparib with metastatic spread to bone and lymph nodes. Liquid CGP after progression on talazoparib detected the truncal *BRCA1* L1620*fs at a VAF of 20%, as well as five additional *BRCA1* mutations: 3 indels, a splice site, and a rearrangement that skips exons 8−9 (VAFs ranged between 0.3 and 2.6%), each predicted to skip the frameshift and restore at least partial *BRCA1* function.

### Case 3
A 67-year-old woman presented with Stage IV ER+/HER2 IHC negative (0−1+) ILC and was treated with letrozole/palbociclib; liquid CGP at progression identified *PIK3CA* E542K mutation and *ERBB2* A775_G776insYVMA. After progression on lapatinib/capecitabine, a metastasis tested ER+/HER2 IHC-positive (3+). The patient was switched to paclitaxel/pertuzumab/trastuzumab, and then gemcitabine/pertuzumab/trastuzumab. A TBx (liver metastasis) and a LBx were collected and profiled during this last treatment. TBx identified mutations in *PIK3CA*, *ERBB2*, and *TBX3*. LBx identified the truncal *PIK3CA* and *ERBB2* mutations as well as potential resistance alterations *FGFR2* K569E (VAF 1.3%), *KRAS* G12V (0.5%), *ERBB2* L755S (0.4%), and *ESR1* D538G (0.1%). The *ESR1* and *ERBB2* L755S mutations are commonly seen after exposure to letrozole and lapatinib, respectively[38,39]. It is worth noting the *ERBB2* exon 20 YVMA insertion may represent resistance to lapatinib at baseline[40,41]. *KRAS* mutations have also recently been described as a form of potential resistance to HER2-targeted therapy[42]. *FGFR2* mutations have recently been described as a mechanism of resistance to hormonal therapy and CDK4/6 inhibitors[43].

Collectively, these cases highlight the role of serial biopsy in identifying tumor evolution, with implications on emerging resistance and therapy selection.

## Discussion
With the rapid development of targeted therapies for breast cancer treatment, circulating tumor DNA profiling has shown potential in identifying actionable alterations, monitoring response, predicting outcome, and assessing treatment evasion[44,45]. The interplay of clonal heterogeneity and tumor evolution is critical for early stages of tumor initiation and progression, as well as late-stage relapse in

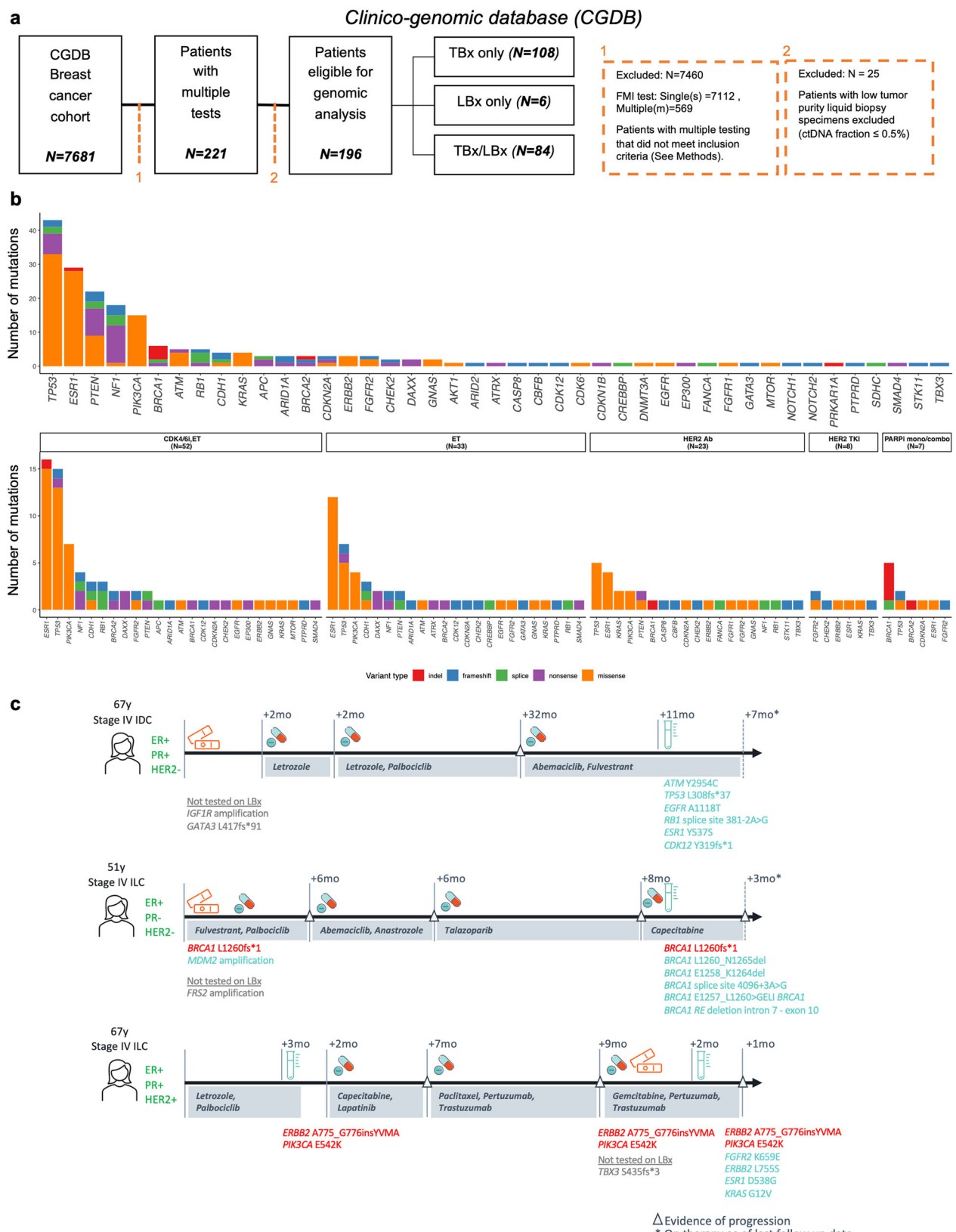

advanced tumors[46]. Liquid biopsy captures both the spatial and temporal clonal heterogeneity in tumors, and can therefore be informative in monitoring treatment-associated response, especially in the metastatic setting. Here, we characterized the baseline and acquired alterations in a large breast cancer cohort comprising

patients who underwent TBx and LBx CGP as part of routine clinical care.

Acquired alterations, potentially associated with resistance, were observed in a majority of patients with matched TBx/LBx, with the highest rates in ER+ disease and those with longer biopsy intervals.

**Fig. 5 | Investigation of patients with longitudinal biopsies in a clinico-genomic database. a** A schematic representing the cohort of 196 patients receiving multiple, longitudinal comprehensive genomic profiling tests, with real-world clinical and outcome data matched from Flatiron Health evaluated in this study. **b** Patterns of acquired short variants (e.g., point mutations, indels) in the overall cohort (top), and based on specific treatment interventions (bottom), revealed treatment-specific patterns of acquired alterations. Acquired alterations (*y*-axis) are colored based on the type of alteration for each gene (*x*-axis). The therapy groups include the following—endocrine therapy (ET) and Cyclin-dependent kinase inhibitors (CDK4/6i), endocrine therapy (ET), HER2-targeted therapy (antibody (Ab)-drug conjugates, tyrosine kinase inhibitors (TKI)), Poly adenosine diphosphate-ribose polymerase inhibitors (PARPi) mono/combo therapy. **c** Three representative patient journeys, displaying the utility of LBx during the course of care, are shown (time shown in months, mo). Shared alterations across biopsies are shown in red. Alterations unique to a specific biopsy are shown in blue while alterations not targeted on other platforms are shown in gray.

Analysis of CGDB revealed that liquid biopsies are used later in the course of patient care, typically in later stage tumors treated with more lines of prior therapy. This observation, in part, likely explains the higher prevalence of acquired alterations and clonal heterogeneity in the liquid biopsy setting. Consistent with prior studies, stratifying patients by intervening treatment, identified known therapy-related associations including acquired *RB1* alterations in the CDK4/6i setting[13], and *BRCA1/2* reversions with PARPi[32,33]. Beyond the well-characterized *ESR1* resistance mutations in the ER+ tumors, MAPK pathway alterations, including *NF1* loss of function alterations and those in RAS pathway receptors (*KRAS, BRAF, NRAS, HRAS*) were also commonly identified in our cohort. Some of these alterations, including *KRAS* G12C, are currently being investigated in solid-tumor basket trials[47], allowing for emerging options for these patients. *BRAF* V600E was recurrently observed as an acquired alteration, notable since the FDA granted accelerated approval for debrafinib/trametinib combination therapy[48,49]. Acquired *PIK3CA* and *ERBB2* mutations, potentially targetable with small molecule inhibitors[1,4] were also common.

Additional rare but actionable alterations that we observed in follow-up LBx may have implications on potential treatment options and combinatorial treatment strategies in these patients. The various FGFR inhibitors in development, taken together with our findings on *FGFR1/2/3* acquired alterations in LBx, might be suggestive of patients with relapsed ER+ breast cancer being an appropriate population for targeted enrollment[43]. Albeit rare, an *ERBB2* fusion identified in a patient with HER2+ breast cancer and an acquired *RET* fusion identified in a patient with an acquired *ESR1*, offer interesting insights into subtype-specific resistance mechanisms and warrant further investigation. These findings highlight that tumor evolution is more of a rule rather than an exception during disease progression, consistent with previous studies showing frequent acquisition of mutations in the resistance setting[9–12,14,21,43].

Tumor heterogeneity, the basis of tumor evolution, has been associated with worse outcome in several tumors[8,50–52]. Thus, understanding what drives heterogeneous adaptation to intervening therapies is critical in devising better therapeutic approaches. Consistent with heterogeneous tumor evolution, acquired LBx alterations in our cohort were frequently subclonal and often exhibited polyclonal patterns. This was particularly true for *BRCA1/2* reversions which were almost universally polyclonal, suggesting convergent evolution of resistance across multiple independent subclones may promote resistance to therapy. Similar findings were seen for *ESR1, PIK3CA,* and *ERBB2*, with patients exhibiting multiple resistance-associated alterations in one or more of these genes. While therapies provide one source of evolutionary pressure, mutations that promote oncogenicity, prevent cell death, or provide a clonal advantage may also be selected for. *TP53* mutations were commonly observed across breast cancer subtypes and may represent a mechanism of selection-driven clonal evolution. While this finding highlights the clinical challenges of therapeutic resistance, these characteristics could aid in the discovery of additional resistance mechanisms, even in the absence of longitudinal or paired data. In a sufficiently sized dataset, alterations frequently observed to be subclonal and/or polyclonal could be flagged as possible resistance alterations and prioritized for additional study and identification of targets for drug discovery. It is also plausible that

the elevated presence of alterations identified in LBx may result from an increased evolutionary burden in certain tumors.

TBx and LBx CGP revealed a high prevalence of actionable alterations, including those in well-characterized drivers such as *BRCA1/2* and *PIK3CA*. However, the extent to which LBx captured the mutational landscape derived from matched TBx profiles was strongly influenced by the estimated tumor fraction, with higher rates in sufficiently shedding samples. LBx CGP with no identified mutations may likely represent low ctDNA shedding; in such cases, reflex testing to TBx, if feasible, may be warranted. This underscores the need for consideration of LBx tumor fraction in the clinical setting where reporting of an accurate estimate of tumor fraction, could enable interpretation of a negative result. Further, the patient-matched paired biopsy analysis in this study was limited to samples profiled on an older LBx technology comprising 62–70 genes, in comparison to the larger TBx panel. As a result, the extent of acquired alterations and consequently, the extent of tumor evolution, may be understated in this cohort. However, the recent approval of larger LBx CGP profiling panels, may help overcome this limitation by providing a broader landscape of the tumor genomics, as observed in our preliminary analysis of 1430 patients who underwent testing on a 324+ gene panel. These tests have the potential to identify a greater breadth of potentially actionable alterations. Notwithstanding these limitations, our study showcased the abundance and diversity of tumor evolution and resistance mechanisms, under selective pressures of routine care and treatment. With the recent FDA-approval of NGS-based LBx CGP assays, and the associated uptake in payor reimbursement, our expectation is that liquid biopsy may grow to be used earlier in the care of breast cancer patients, potentially to enable tumor profiling at initial recurrence.

A limitation of this study was the lack of matched white blood cell data. Mutations that are rare in solid tumors such as *JAK2* V617F can be assumed to be non-tumor derived and their presence may even be a reason for clinical follow-up and monitoring for development of myelodysplasia. Mutations that are more common in solid tumors and considered biomarkers for certain therapies pose more of a problem. Mutations in *ATM, CHEK2* have been observed in white blood cells[37]. CH variants have even been observed in *BRCA2* and *KRAS*, albeit rarely and not in patients with breast cancer[53,54]. Somatic mutations with low VAFs thus need to be interpreted with some level of caution. It is worth noting that PARPi treatment is not currently recommended by any regulatory agency guidelines for breast cancer with *ATM2* and *CHEK2*. Our study is also limited based on the clinical and molecular (e.g., ER status) available in the CGP cohort. While acquired alterations are observed in paired-sample biopsies, especially those with longer biopsy intervals, the tumor features and clinical interventions are unknown for most of these cases.

Collectively, findings from our study reveal the dynamic evolution of breast tumors over time. Since both baseline and acquired alterations are potentially targetable (e.g., activating *PIK3CA, ERBB2, KRAS, BRAF, FGFR1/2/3*), longitudinal profiling throughout the course of care may provide additional treatment options, through combinatorial therapies or switching therapy regimens based on the evolving tumor landscape, to improve outcomes for patients with breast cancer. With poor outcomes in the relapsed and metastatic setting, clinical trials are urgently needed to assess whether targeting these alterations can

improve outcomes. This could be achieved through an umbrella trial in the relapsed setting, matching patients to targeted therapies. Similar approaches have been taken for the CUPISCO and TAPISTRY trials which match patients to targeted therapy arms based on their genomic profiles[55,56]. Our study highlights the potential for liquid biopsy approaches to better characterize the evolutionary landscape, guide clinical decision making, monitor treatment response/resistance and ultimately aid in personalized care for patients with breast cancer.

## Methods
Approval for this study, including a waiver of informed consent and Health Insurance Portability and Accountability Act waiver of authorization, was obtained from the Western Institutional Review Board (Protocol #20152817). The Institutional Review Board granted a waiver of informed consent under 45 CFR § 46.116 based on review and determination that this research meets the following requirements: (i) the research involves no more than minimal risk to the subjects; (ii) the research could not practically be carried out without the requested waiver; (iii) the waiver will not adversely affect the rights and welfare of the subjects. For the clinico-genomic database (CGDB), IRB approval of the study protocol was obtained prior to study conduct and included a waiver of informed consent based on the observational, non-interventional nature of the study (WCG IRB, Protocol No. 420180044). No compensation was provided as samples were obtained during routine clinical care.

### Comprehensive genomic profiling
Comprehensive genomic profiling (CGP) of formalin-fixed, paraffin embedded tissue biopsy sections (TBx) from 29,704 breast cancer patients was performed using FoundationOne®/FoundationOne® CDx. Blood-based CGP (LBx) of 3339 patients was performed using FoundationACT®/FoundationOne® Liquid and 712 patients were profiled on both platforms (LBx on or after TBx). Hybrid capture was carried out on at least 324 cancer-related genes and select introns from up to 31 genes frequently rearranged in cancer in our TBx assay, while the LBx assay covered up to 70 cancer-related genes and select introns from up to seven genes to identify short variants (base substitutions and indels), copy number alterations, and rearrangement events (Supplementary Data 2)[57,58]. Further, comparisons between TBx and LBx were limited to the genomic regions covered in the TBx and LBx assays within these 70 genes. Patient populations represented all-comers sequenced during routine clinical care. Well-characterized germline alterations within BRCA1/2[59], were grouped together as founder mutations. Receptor subtype was available for a subset of cases. An independent assessment of 1430 patients profiled using FoundationOne® Liquid CDx[22] was also performed.

Sequence analysis methods and validation of the CGP platform used in this study have been described previously by Frampton and colleagues (https://www.accessdata.fda.gov/cdrh_docs/pdf17/P170019S006B.pdf)[22,57]. For tissue biopsy analysis, base substitution detection was performed using a Bayesian methodology, which enables the detection of somatic mutations at low mutant allele frequency (MAF) and increased sensitivity for mutations at hot-spot sites through the incorporation of tissue-specific prior expectations. Reads with mapping quality <25 were discarded, as were base calls with quality ≤2. Final calls were made at MAF of ≥5% (MAF ≥1% at hot spots) to avoid false-positive calls, after filtering for strand bias (Fisher test, $p < .001$), read location bias (Kolmogorov–Smirnov test, $p < .001$), and presence in ≥two normal controls. To detect short insertions or clinico-genomic databaseThis study also utilized a US-based de-identified Flatiron Health-Foundation Medicine clinico-genomic database (FH-FMI CGDB), comprising approximately 280 cancer clinics (~800 sites of care). Retrospective longitudinal clinical data were derived from electronic health records (EHR), comprising patient-

level structured and unstructured data, curated via technology-enabled abstraction, and were linked to genomic data derived from FMI CGP tests in the CGDB by de-identified, deterministic matching[60].The study included 7681 patients satisfying the following cohort inclusion criteria: (1) Chart-confirmed diagnosis of breast cancer (data collected through September 30, 2020), (2) Had at least two documented clinical visits in the FH network on or after January 1, 2011, (3) Underwent CGP testing on a pathologist-confirmed breast cancer tumor specimen, at FMI, on or after date of chart-confirmed initial diagnosis of breast cancer, on a sample collected no earlier than 30 days before the FH diagnosis date. A total of 6757 patients received at least one TBx test and a total of 1150 patients received at least one LBx test (Fig. 2); these include a subset of patients who received both TBx and LBx testing.Amongst 1852 patients identified to be ER positive or PR positive and HER2 negative at metastatic diagnosis, 1083 received a combination of CDK4/6 inhibitor treatment and endocrine therapy in the metastatic non-maintenance setting; 713 patients had pre-treatment biopsies (specimens collected prior to start of treatment) and 414 had post-treatment biopsies (specimens collected post 90 days after the start of treatment). Amongst 788 patients identified to be HER2 positive at metastatic diagnosis, 304 received a combination of HER2 antibody (Ab)-based targeted therapy and chemotherapy in the metastatic non-maintenance setting (192 patients had pre-treatment biopsies and 144 had post-treatment biopsies) and 107 received a combination of HER2 TKI and chemotherapy in the metastatic non-maintenance setting (73 patients had pre-treatment biopsies and 39 had post-treatment biopsies). And finally, amongst 179 patients who received PARP inhibitors, as monotherapy or combination non-maintenance therapy in the metastatic setting, 162 patients had pre-treatment biopsies and 23 had post-treatment biopsies.For the above-described analysis, if multiple specimens were extracted from patients for the purpose of CGP testing and or patients had received multiple lines of therapy-in-question, the longest specimen collection date-therapy start date combination was chosen per patient. Prevalence of only genomic short variants (e.g., single base substitutions, short insertions/deletions, splice alterations) were compared between the pre- and post-treatment biopsies and reported.Paired samples were considered if they were collected at least 30 days apart, the patient was in network during both the CGP tests and liquid biopsy specimens (where relevant) had an estimated tumor fraction greater than 0.5%. After applying these filters, a total of 196 patients with longitudinal samples were available.

### Estimation of the ctDNA fraction and clonality in liquid biopsy samples
An estimation of the ctDNA fraction in liquid biopsy samples was performed using two complementary methods: (i) a tumor fraction estimator (TFE), based on a measure of tumor aneuploidy, and (ii) the maximum somatic allele frequency (MSAF) method, using allele fraction from somatic coding alterations to estimate ctDNA fraction[61]. For short variants (base substitutions and indels), clonal fraction of a variant was calculated as the ratio of the variant allele fraction (VAF) to the sample estimated ctDNA fraction.

### Comparison of TBx and LBx profiles
Comparisons of gene alteration prevalence between the platforms were limited to the 70 genes captured in the LBx assay (Supplementary Data 2). For comparisons based on site of biopsy, the tissue biopsy cohort was divided into those profiled from breast (local, $n = 9860$) and other metastatic sites ($n = 13,811$), with each being compared against the full LBx cohort ($n = 3339$). For comparisons based on receptor subtype, additional steps were applied in pre-processing the data. Liquid biopsy samples in our cohort do not have receptor subtype information, and therefore, the status from the matched tissue biopsy sample was utilized, where available. Further, to maximize the number of samples for this analysis, for patients with both tissue and

liquid biopsy profiling available, the liquid biopsy sample was prioritized to be selected. Each patient is included only in one of the platforms (TBx, LBx) being compared.

For the paired LBx/TBx cohort, percent positive agreement was determined at the variant level, by matching on coding and protein effect for all identified short variants, using TBx as the reference. To validate findings from the paired LBx/TBx cohort, a similar analysis of short variants identified within these 70 genes in 1328 patients with longitudinal TBx was performed.

Further, we also inspected the mapped bam files for the matched tumor tissue to see if there was sub-threshold evidence for the single base substitutions detected only in the matched liquid biopsy sample using *pysam pileup* ($n = 381$ variants).

## Statistical methods

Differences in prevalence of gene alterations between TBx and LBx assays, as well as patterns of co-occurrence and mutual exclusivity between gene alterations were tested using a Fisher's exact test. Two-sided $P$-values were calculated for each comparison and then adjusted for multiple hypothesis testing using the Benjamini–Hochberg FDR method. For continuous variables (e.g., clonal fraction, LBx tumor fraction), Wilcoxon rank sum test was used to test for differences between specific groups; two-sided $P$-values were calculated for each comparison. For all analyses, the significance level was set to 0.05. Statistics, computation, and plotting were carried out using Python 2.7 (Python Software Foundation) and R 3.6.1 (R Foundation for Statistical Computing).

## Reporting summary

Further information on research design is available in the Nature Portfolio Reporting Summary linked to this article.

## Data availability

The sequencing data generated in this study are derived from clinical samples. The data supporting the findings of this study are provided within the paper and its supplementary files. All supplementary information accompanying the different analyses and figures presented in this study are provided in the Supplementary Data files. Due to HIPAA requirements, we are not consented to share individualized patient genomic data, which contains potentially identifying or sensitive patient information. Foundation Medicine is committed to collaborative data analysis, and we have well-established, and widely utilized mechanisms by which investigators can query our core genomic database of >600,000 de-identified sequenced cancers to obtain aggregated datasets. Requests for collaborative datashares can be made by contacting the corresponding author(s) and filling out a study review committee form. Once approved, investigators are required to sign a data transfer agreement. Written proposals are considered at monthly meetings and data transfer agreements expire 18 months from execution of the agreement.

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

## Author contributions

S.S. and E.S.S. conceptualized the study, interpreted the results, and wrote the manuscript. S.S., D.X.J., H.T., K.M., D.P., and E.S.S. performed the data analyses. N.D., R.H., and K.M. assisted with data collection. S.S. performed data visualization and curation. E.S.S. assisted with project supervision and administration. All authors, including O.G., J.S.R., J.C., B.D., L.D., G.M.F., L.M., S.O., J.M.V., G.R.O. and P.S.H. contributed to detailed manuscript revision and review.

## Competing interests

S.S., D.X.J., H.T., K.M., N.D., D.P., O.G., J.S.R., R.H., B.D., L.D., G.M.F., P.S.H., J.M.V., G.R.O., and E.S.S. are employees at Foundation Medicine, Inc., with an equity interest in Roche. L.M. is an employee at Genentech, Inc, with an equity interest in Roche. K.M. had employment with Foundation Medicine Inc., and Oncocyte, with an equity interest in Roche and Oncocyte. J.C. had employment with Foundation Medicine Inc., and GlaxoSmithKline, with an equity interest in Roche and GlaxoSmithKline. S.O. has no potential conflicts of interest to declare within the scope of this work.
