## [Peer Review File · Nature Communications]

REVIEWER COMMENTS

Reviewer #1 (Remarks to the Author): Expert in breast cancer genomics, intratumour heterogeneity, and cancer evolution

- What are the noteworthy results?

The work has analysed results from an impressively large number of breast cancer patients, both tumor biopsies and liquid biopsies. The most noteworthy results are the frequencies of alterations in cancer related genes, both in tumors and in later liquid biopsies. The findings are confirming earlier works (as the authors discuss), but also find more rare alterations, in particular in the liquid samples taken later in the course of the disease.

- Will the work be of significance to the field and related fields? How does it compare to the established literature? If the work is not original, please provide relevant references.

See above; it adds knowledge about frequencies etc. The major challenge is the clinical information, there are very limited details of patient selection criteria (method section) and that it is not designed to elucidate types of evolution (see below).

The information about included patients are confusing, Methods section start with a section about 29704 breast cancer patients, then a later section defines 7681 metastatic breast cancer patients. The latter number is not the same as in Table 1. This table also show that the “metastatic” cohort have many patients with early stage at time of ordering test, this is confusing. It also has patients with stage zero (0)? The first cohort seems to be a mix of both early stage and advanced/metastatic disease. There is a major reduction in number of analysed cases when stratified into clinically relevant groups, for instance by treatment history (Table S13).

- Does the work support the conclusions and claims, or is additional evidence needed?

The work claims to “we characterized the clonal evolution landscape“, which might be an overstatement. There is no thorough analysis of the primary tumor by multiple sampling and analyses, no characterization of metastases. The statement should be saved for more comprehensive studies designed to elucidate tumor evolution, in the present work the frequencies of alterations are the most interesting.

Along the same line, it also concludes that it “reveals mechanisms of convergent tumor evolution” (title). Commonly, convergent evolution is defined as a situation where two independent lineages in the tumor mutate the same driver gene, leading to independent clonal expansions (see for instance PMID: 28110020). The data presented cannot support this.

The introduction also state that «In this study, we [...] to characterize the diversity of acquired mutations, their association with therapeutic interventions, as well as mechanisms of tumor evolution and relapse,...]. They do perform the association study, but there is no evidence that this study reveals a

mechanistic understanding of evolution and no data where connection between evolution and relapse can be studied.

The introduction is very short.

- Are there any flaws in the data analysis, interpretation and conclusions? Do these prohibit publication or require revision?

The results are mainly frequencies and no advanced statistics are used. Are corrections for multiple testing used? See above for concerns about the conclusions.

- Is the methodology sound? Does the work meet the expected standards in your field?

The methodology (molecular testing and bioinformatics) is based on “black box analyses” provided by a commercial partner. There are no novel algorithms/bioinformatics in the paper. The data can therefore not be checked by using other annotation/informatics pipelines etc.

There is no information about ethical approval of the study including patient consent.

- Is there enough detail provided in the methods for the work to be reproduced?

The data can probably be reproduced for breast cancer patients in general, but not for clinically relevant groups, as the descriptive information of the clinical features of the patient cohort is very limited, and a large number of patients have important information missing.

Reviewer #2 (Remarks to the Author): Expert in breast cancer genomics and liquid biopsies

This is a large-scale genomic study of patients with breast cancer who underwent clinical tumor sequencing or liquid biopsy utilizing well-validated Foundation Medicine’s NGS assays. The study includes a total of 29,704 tumors and 3,339 liquid biopsies. Clinical data was collected by Flatiron Health and was available for almost a third of the patients (6,757 patients with tissue sequencing and 1,150 patients with liquid biopsy).

The authors report high concordance between ctDNA and tissue sequencing, particularly in the samples with high ctDNA fraction. The mutational landscapes of ctDNA and tumor tissue were mainly comparable. However, the frequency of alterations known to be associated with resistance to targeted therapies (e.g. ESR1 mutations, alterations involving the MAPK pathway, as well as RB1 and PTEN loss of function mutations) were higher in ctDNA than tumor consistent with the notion that such resistant mechanisms are often subclonal and the tumor tissue biopsy may not capture them due to tumor spatial heterogeneity. The authors also describe detection of reversion mutations in ctDNA samples of gBRCA1/2 carriers following exposure to PARPi, highlighting the utility of ctDNA in detection of such

variants. The study also underlines the high prevalence of convergent evolution resulting in polyclonal resistance and provides clinical rationale for utilization of liquid biopsy assays to monitor tumor evolution and to identify potential actionable alteration that can emerge throughout the course of the disease.

Overall this is well-written and timely manuscript on the understanding of practical use of ctDNA and tissue NGS for detection of actionable alterations and monitoring tumor evolution in breast cancer. By and large, the study is robust and of great practical interest and covers an important topic which is highly relevant to management of breast cancer. Perhaps the main limitation of this manuscript is that the authors seemed not to have harnessed the full potentials of their large genomic and clinical data. Even though almost 30k patients have undergone tumor or ctDNA profiling, the main analyses are based on a very small subset of these patients. For example, out of 7,681 patients with matched clinical data, only 196 were included in the clinico-genomic study, limiting the power of such analyses for identifying novel results. Indeed, the majority of the findings of this study have been previously described elsewhere, thus limiting the novelty of this study.

I have a few questions and comments:

- Table 1: Histology and receptor subtype are only reported for the 712 paired cases. This data should be reported for the full cohort. What are the sites of biopsies for the 29K tumor tissue samples?

- Figure 2A-2B: The authors report higher frequency of alterations associated with resistance to targeted therapies in ctDNA than tumor tissue. It is important to present this analysis by breast cancer (Similar to Fig S4 for mets vs primary).

- Figure 2C: What was the rate of multiple PIK3CA mutation in ctDNA and tissue? Vasan et al (Science 2019) identified distinct patterns of multiple PIK3CA mutations in cis that result in higher PI3K activity and increased sensitivity to PI3K inhibitors. Do the authors observe similar patterns in this large genomic data both in the tumor and ctDNA?

- ctDNA-tissue concordance analysis: The ctDNA assays utilized in this study have very limited utility in early-stage breast cancer. Hence, I would suggest limiting the tissue-ctDNA concordance analysis for the somatic variants to patients who were metastatic at the time of liquid biopsy collection. This will make the concordance results more realistic. In addition to Fig S5 and Table S7, this data can also be presented by receptor subtype.

- Figure 3A, the authors report multiple ctDNA-only variants that were not called in the tumor. How many of these variants had supporting reads in the matched tumor tissue but were not called as they were below the calling thresholds for the tissue assay?

- Line 152: The authors report that “germline alterations were detected across a wide range of tumor fractions, as expected since the former can be detected in non-tumor DNA even when ctDNA shed is low”. This is an obvious statement as the germline mutation calling is indeed expected to be more accurate when ctDNA fraction is low. The challenge for germline calling using the ctDNA-only approach is when the tumor fraction is high. The authors should report the accuracy of germline calling in high fraction ctDNA samples.

- Figure S5C: Seems like this plot includes both germline or somatic BRCA1/2 mutations. I would suggest presenting data separately for germline and somatic variant in the same plot.

- Line 175-181: The authors report identifying acquired amplification and rearrangements. Detection of large structural variants in ctDNA can often be challenging. Were these variants identified mainly in samples with high ctDNA fraction?

- Line 183: The authors report multiple acquired PIK3CA mutations in ctDNA. The mutations appear to be the ones associated with APOBEC mutagenesis. What was the mutational burden for these ctDNA samples? It would be informative to perform mutational signature analysis on the tumor and/or ctDNA samples of these cases to assess for APOBEC associated mutational signatures 2 and 13.

- Figure 3D and lines 204-218:

1) It is important to highlight the performance of ctDNA in detecting the BRCA1/2 reversion variants as compared to the tumor and explicitly mention what fraction of these variants were also observed in the tumor or not.

2) What were the rates and patterns of BRCA1/2 reversion mutations in the full ctDNA cohort (3,339 samples)?

- Line 220: The authors report that “Consistent with tumor heterogeneity and clonal evolution dynamics, acquired short variants in LBx revealed a lower clonal fraction relative to shared alterations with TBx”. The methods described in Tukachinsky et al and this manuscript, do not consider the copy number state of the variant and may over or underestimate the clonality of the ctDNA-only mutations. Would it be feasible to include the tumor-derived allele specific copy numbers in this analysis?

- Figure 3F: This is an important finding and highlights the rate of p[olyclonal resistance in breast cancer. What are the patterns of multiple mutations for these alterations in the full ctDNA cohort stratified by receptor subtype (n=3,339)?

- Lines 238-259: As the authors indicated, CH can potentially be a major source of somatic mutation in cfDNA. In addition to TP53 and JAK2 that are mentioned here, CH also frequently affects ATM, NF1, CHEK2 in cancer patients and can be detected in cfDNA even in young individuals. These alterations are reported to be enriched in ctDNA (Fig 2B) and are often reported to be seen only in ctDNA (Fig 3A). How confident are the authors that these alterations are indeed tumor-derived and are not arising from CH. Are matched buffy coats available for assessment of CH for these patients? Additionally, non-recurrent CH mutations may affect non-canonical CH genes (Chabon et al, Nature 2020). BRCA1 and BRCA2 are large genes and some of their ctDNA-only non-pathogenic mutations could be passenger CH mutations. This issue needs to be further explored and discussed.

- CGDB analysis: It is not clear why the analysis of the CDGB cohort was primarily limited to only the patients with multiple tests, excluding >97% of the patients. The current analysis based on small numbers (196 patients: 104 with paired tumor tissues and 84 with paired tissue and ctDNA) provides some anecdotal evidence for enrichment of known mechanisms of resistance to certain therapies (MAPK alterations and ESR1 mut in 33 patients post-endocrine therapy, RB1 in 52 patients post-CDK4/6i and ERBB2 mut and PIK3CA mut in 23 patients after anti-HER2 therapy). The clinico-genomic analysis of the full cohort (1,150 patients with ctDNA data after excluding the cases with low purity and the 6,757 patients with tissue data) could have potentially provided robust associations with prior exposure to different classes of targeted therapies (endocrine therapy with AIs vs SERD vs SERm, CDK4/6i, PI3Ki, anti-HER2 Abs and TKIs, IO, etc) and shed light on potential novel mechanisms of resistance. The current analysis of the paired samples can be reported as a subset analysis of the full cohort to provide a rationale for serial ctDNA monitoring.

Case #1: ATM VAF is highly discordant with the rest of the alterations identified in the ctDNA and raises the possibility that this was potentially a CH variant. How does the authors explain this discordance?

Minor comments

- The number of samples and patients included in each analysis are not consistently reported in the legends and the text.

- The bar plots have the X-axis labels on top of the plot. It would be easier to follow the figure if these labels are placed below the X-axis line.

- Figure S2: It would be informative to add the percentages for TBx and LBx to the heatmap. The heatmap is also hard to follow. The colors are very close, and a large number of variants are coded as "multiple variants". I would suggest clarifying the colors or just use a standard oncoprint to presents the genomic landscape.

- Line 105- 5% seems to be a typo and it should be 0.5%.

- Line 113: The short variant are defined in the methods. Please also define it in the text.

- Line 186: Would mention "qualified for PI3K inhibitors" rather than "alpelisib-qualifying context".

- Line 211: BRCA should be BRCA2

Thank you for the invitation to submit a major revision of our manuscript. We have made substantial revisions to the manuscript and have addressed all the reviewers' concerns. Please find our point-by-point response to the reviewer comments below:

Reviewer 1:

• What are the noteworthy results?

The work has analysed results from an impressively large number of breast cancer patients, both tumor biopsies and liquid biopsies. The most noteworthy results are the frequencies of alterations in cancer related genes, both in tumors and in later liquid biopsies. The findings are confirming earlier works (as the authors discuss), but also find more rare alterations, in particular in the liquid samples taken later in the course of the disease.

We thank the reviewer for their kind words and for acknowledging the power of our large dataset.

• Will the work be of significance to the field and related fields? How does it compare to the established literature? If the work is not original, please provide relevant references. See above; it adds knowledge about frequencies etc. The major challenge is the clinical information, there are very limited details of patient selection criteria (method section) and that it is not designed to elucidate types of evolution (see below).

The information about included patients are confusing, Methods section start with a section about 29704 breast cancer patients, then a later section defines 7681 metastatic breast cancer patients. The latter number is not the same as in Table 1. This table also show that the "metastatic" cohort have many patients with early stage at time of ordering test, this is confusing. It also has patients with stage zero (0)? The first cohort seems to be a mix of both early stage and advanced/metastatic disease. There is a major reduction in number of analysed cases when stratified into clinically relevant groups, for instance by treatment history (Table S13).

- 1) We have added additional details on the clinical inclusion criteria to clarify that these cohorts represent an all-comers population of those who have received CGP; we agree that a selected cohort would have less generalizability.
- 2) With regards to the cohort sizes (29,704 v 7681), 29,704 is the size of the genomics cohort without clinical treatment data; the 7681 represent patients in the Flatiron Health database with full clinical treatment histories. The latter enables treatment-genomic associations and information on lines of therapy, practice type, and ECOG. To make it clear that these analyses were run on separate datasets, we have made a few changes to the text: A) we have moved the section describing the CGDB dataset immediately adjacent to the CGP section in the methods. B) We have altered the language of the CGDB section to make it clear that this is a distinct analysis. We now say that "this study also utilized a US-based de-identified Flatiron Health-Foundation Medicine clinico-genomic database" rather than "This study used US-based de-identified Flatiron Health-Foundation Medicine clinico-genomic database". Additionally, the numbers presented in Table 1, represent patients who received at least one TBx test (n=6,757) and those who received at least one LBx test (n=1,150) for the analysis of each cohort; these include a subset of patients who received both TBx and LBx testing. We have also clarified this in Table 1 and in the Methods section titled "Investigation of breast cancer cohort in the clinico-genomic database", in the revised manuscript.

- 3) With Regards to the stage 0 patients in the clinical cohort, we agree that this is confusing as described in the methods. While the main text accurately reflects the clinico-genomic cohort as all-comers with breast cancer, the methods incorrectly described an inclusion criteria of “Chart-confirmed diagnosis of metastatic breast cancer”. This is incorrect and should have stated “Chart-confirmed diagnosis of breast cancer”. This error was introduced by using old methods (pre-2020) when the clinico-genomic database was limited only to metastatic samples. We apologize for this error and have updated the methods. While the cohort is largely metastatic, there are some patients with physician-reported early-stage disease who have received CGP testing including 0.4% of patients with reported stage 0 disease (DCIS/LCIS) sent for testing.
- 4) With regards to Table S13 (now Table S15), the reduction in patient numbers is a result of limiting to patients with paired samples (pre- and post- therapy). To address the reviewer’s concern, we have generated an additional set of analyses in the overall cohort, examining the prevalence of alterations in patient exposed or not exposed to a given therapy class: CDK4/6 inhibitor treatment and endocrine therapy (N=713 pre-therapy, N=414 post-therapy), HER2 hormone therapy and chemotherapy (N=224 pre-therapy, N=158 post-therapy), PARP inhibitor therapy (N=162 pre-therapy, N=23 post-therapy). These results have been presented in Figure S13 and Table S17. The Methods section has been updated to reflect these changes. Based on the increased number of samples, we believe this provides additional value beyond the 196 paired samples analyzed previously. This analysis identified similar findings, with associations of *ESR1* and *RB1* alterations in patients receiving a combination of endocrine therapy and CDK4/6i therapy, and *BRCA1/2* reversions with PARPi therapy, among other associations.

• Does the work support the conclusions and claims, or is additional evidence needed?

The work claims to “we characterized the clonal evolution landscape“, which might be an overstatement. There is no thorough analysis of the primary tumor by multiple sampling and analyses, no characterization of metastases. The statement should be saved for more comprehensive studies designed to elucidate tumor evolution, in the present work the frequencies of alterations are the most interesting.

Along the same line, it also concludes that it “reveals mechanisms of convergent tumor evolution” (title). Commonly, convergent evolution is defined as a situation where two independent lineages in the tumor mutate the same driver gene, leading to independent clonal expansions (see for instance PMID: 28110020). The data presented cannot support this.

The introduction also state that «In this study, we [...] to characterize the diversity of acquired mutations, their association with therapeutic interventions, as well as mechanisms of tumor evolution and relapse,...]. They do perform the association study, but there is no evidence that this study reveals a mechanistic understanding of evolution and no data where connection between evolution and relapse can be studied.

The introduction is very short.

- 1) We appreciate the reviewer’s comments on clonal evolution. We agree that this term encompasses a number of clinical scenarios and detection methods (including sequencing of multiple spatial regions in the primary tumor and metastases). To avoid confusion or mis-interpretation of our work, we modified the final paragraph of the

introduction to remove notes on our study providing mechanistic insight into clonal evolution. Specifically, we completely removed the clause “as well as mechanisms of tumor evolution and relapse, in a real-world setting” in the introduction. Similarly, in the discussion we modified the clause “Here, we characterized the tumor evolutionary landscape in a large breast cancer cohort comprising patients who underwent TBx and LBx CGP as part of routine clinical care.” to “Here, we characterized the baseline and acquired alterations in a large breast cancer cohort comprising patients who underwent TBx and LBx CGP as part of routine clinical care.”

- 2) Regarding “convergent tumor evolution”, we believe we have demonstrated this in our manuscript. Figures 3F-H highlight that the acquired alterations are frequently subclonal and polyclonal. Figure 3F shows as many as 22 subclonal BRCA1/2 reversion alterations suggestive of convergent evolution to become PARPi resistant. Similarly, figures 3G and 3H show that most of the acquired alterations are subclonal and are often polyclonal in *PIK3CA*, *ESR1*, and *RB1*. The subclonality and polyclonality is consistent with “independent lineages in the tumor mutating the same driver/resistance gene”. Is there additional data that the reviewer would like to see to support this?
- 3) Regarding the description of our study as providing mechanistic insight, we have removed this clause in its entirety from the introduction (see pt 1)
- 4) The introduction has been expanded by two paragraphs to provide more background and rationale for the study

• Are there any flaws in the data analysis, interpretation and conclusions? Do these prohibit publication or require revision?

The results are mainly frequencies and no advanced statistics are used. Are corrections for multiple testing used? See above for concerns about the conclusions.

We thank the reviewer for raising concerns about the statistics used in this study. Although analyses were corrected for multiple hypothesis testing, the figure legends were not always clear (for example, figure 2A makes no note about FDR: “(A) Prevalence of the 20 most frequently altered genes (Fisher’s exact test; p values thresholds *: 0.05, **: 0.01, ***: 0.001).”). We have modified the legends and methods text to be clear about FDR corrections. In addition, we have aggregated the statistical methods into their own methods section (“Statistical Methods”) for clarity.

• Is the methodology sound? Does the work meet the expected standards in your field?

The methodology (molecular testing and bioinformatics) is based on “black box analyses” provided by a commercial partner. There are no novel algorithms/bioinformatics in the paper. The data can therefore not be checked by using other annotation/informatics pipelines etc. There is no information about ethical approval of the study including patient consent.

- 1) We appreciate the reviewer’s feedback. While raw data cannot be released due to patient compliance rules, we have expanded the description of the methods, providing additional details on how the calls are made. This new paragraph can be found at the end of the section “Comprehensive genomic profiling” in the Methods section.
- 2) Regarding consent, our IRB approval statement was inadvertently omitted in the first draft. We have added this back in. We now write: “Approval for this study, including a waiver of informed consent and Health Insurance Portability and Accountability Act

waiver of authorization, was obtained from the Western Institutional Review Board (protocol #20152817).”

**• Is there enough detail provided in the methods for the work to be reproduced?
The data can probably be reproduced for breast cancer patients in general, but not for clinically relevant groups, as the descriptive information of the clinical features of the patient cohort is very limited, and a large number of patients have important information missing.**

We thank the reviewer for these comments. We have added additional data on clinically relevant receptor subgroups in Table S1, Figure S1, Table S4, Figure S6, Table S9 and Figure S11. While there is little missingness in the cohort with full clinical information, we have added a discussion of this limitation in the discussion of the CGP cohort. We write: “Our study is also limited based on the clinical and molecular (e.g., ER status) available in the CGP cohort. While acquired alterations are observed in paired-sample biopsies, especially those with longer biopsy intervals, the tumor features and clinical interventions are unknown for most of these cases.”

Reviewer 2:

This is a large-scale genomic study of patients with breast cancer who underwent clinical tumor sequencing or liquid biopsy utilizing well-validated Foundation Medicine’s NGS assays. The study includes a total of 29,704 tumors and 3,339 liquid biopsies. Clinical data was collected by Flatiron Health and was available for almost a third of the patients (6,757 patients with tissue sequencing and 1,150 patients with liquid biopsy).

The authors report high concordance between ctDNA and tissue sequencing, particularly in the samples with high ctDNA fraction. The mutational landscapes of ctDNA and tumor tissue were mainly comparable. However, the frequency of alterations known to be associated with resistance to targeted therapies (e.g. ESR1 mutations, alterations involving the MAPK pathway, as well as RB1 and PTEN loss of function mutations) were higher in ctDNA than tumor consistent with the notion that such resistant mechanisms are often subclonal and the tumor tissue biopsy may not capture them due to tumor spatial heterogeneity. The authors also describe detection of reversion mutations in ctDNA samples of gBRCA1/2 carriers following exposure to PARPi, highlighting the utility of ctDNA in detection of such variants. The study also underlines the high prevalence of convergent evolution resulting in polyclonal resistance and provides clinical rationale for utilization of liquid biopsy assays to monitor tumor evolution and to identify potential actionable alteration that can emerge throughout the course of the disease.

Overall this is well-written and timely manuscript on the understanding of practical use of ctDNA and tissue NGS for detection of actionable alterations and monitoring tumor evolution in breast cancer. By and large, the study is robust and of great practical interest and covers an important topic which is highly relevant to management of breast cancer. Perhaps the main limitation of this manuscript is that the authors seemed not to have harnessed the full potentials of their large genomic and clinical data. Even though almost 30k patients have undergone tumor or ctDNA profiling, the main analyses are based on a very small subset of these patients. For example, out of 7,681 patients with matched clinical data, only 196 were included in the clinico-genomic study, limiting the power of such analyses for identifying novel results. Indeed, the majority of the findings

of this study have been previously described elsewhere, thus limiting the novelty of this study.

- 1) We thank the reviewer for their summary of our manuscript and its strengths.
- 2) Regarding the note about using only a subset of the 7,681 patients, we have generated an additional set of analyses in the total cohort (Figure S13 and Table S17), examining the prevalence of alterations in patients exposed or not exposed to a given therapy class: CDK4/6 inhibitor treatment and endocrine therapy (N=713 pre-therapy, N=414 post-therapy), HER2 hormone therapy and chemotherapy (N=224 pre-therapy, N=158 post-therapy), PARP inhibitor therapy (N=162 pre-therapy, N=23 post-therapy). Based on the increased number of samples, we believe this provides additional value beyond the 196 paired samples. Similar findings were observed with rates of *ESR1*, *RB1* higher post endocrine therapy+CDK4/6i therapy and *BRCA1/2* reversions following PARPi therapy.
- 3) Regarding novelty, we agree that many of the high frequency events have been reported before (e.g., *ESR1*, *NF1*, *RB1*, *BRCA* reversions). However, we observed an interesting longtail of acquired alterations with potential clinical relevance/actionability including *FGFR*, *RET*, and *ERBB2* fusions, PD-L1 amplifications, RAS alterations, including *KRAS* G12C, and *CDK12* rearrangements. These findings are included in the supplement but are not as prominent in the main figures. To better highlight these, we have modified the abstract to note that we observe a diversity of rare but potentially actionable acquired alterations and have also moved the acquired copy number and rearrangement events to the main figure (Figure 3D and 3E).

- Table 1: Histology and receptor subtype are only reported for the 712 paired cases. This data should be reported for the full cohort. What are the sites of biopsies for the 29K tumor tissue samples?

We agree that providing information on the full cohort is valuable. We have added supplemental table that reports the histology, receptor subtype, and biopsy site for the 29K tumor tissue samples (Table S1). As expected for an all-comers population, most patients had ER+ disease and were histologically invasive ductal carcinomas.

- Figure 2A-2B: The authors report higher frequency of alterations associated with resistance to targeted therapies in ctDNA than tumor tissue. It is important to present this analysis by breast cancer (Similar to Fig S4 for mets vs primary).

We thank the reviewer for their suggestion. We have added supplemental Figure S1 and Table S4, reporting the frequency of alterations in tissue versus liquid biopsies, broken down by subtype. Since we do not have receptor subtype information for our liquid biopsy cohort, this analysis required additional steps in pre-processing the data. Briefly, to annotate liquid biopsy samples in our cohort, we utilized the status from the matched tissue biopsy sample, where available. Therefore, to ensure that each patient is included only in one of the platforms being compared, the liquid biopsy was prioritized to be picked for the patient, if both liquid and tissue biopsies were available. These additional details have been added to the Methods section of the paper (page 15).

- Figure 2C: What was the rate of multiple PIK3CA mutation in ctDNA and tissue? Vasan et al (Science 2019) identified distinct patterns of multiple PIK3CA mutations in cis that result in higher PI3K activity and increased sensitivity to PI3K inhibitors. Do the authors

observe similar patterns in this large genomic data both in the tumor and ctDNA?

We have added the rates of multi-PIK3CA alteration to the revised manuscript text (page 4). We find 4.5% of tissue samples and 7.2% of liquid samples harbor 2 or more PIK3CA alterations, possibly sensitizing to PI3K inhibitors.

- ctDNA-tissue concordance analysis: The ctDNA assays utilized in this study have very limited utility in early-stage breast cancer. Hence, I would suggest limiting the tissue-ctDNA concordance analysis for the somatic variants to patients who were metastatic at the time of liquid biopsy collection. This will make the concordance results more realistic. In addition to Fig S5 and Table S7, this data can also be presented by receptor subtype.

We agree with the reviewer that ctDNA assays have typically been utilized in more advanced stages. Unfortunately, we do not have resolution into the stage at the time of liquid biopsy. Our assays have largely been utilized in advanced stages for both tissue and liquid biopsy profiling, so we believe it is most likely representative of advanced stage breast cancer profiles. For tissue biopsy-based profiling, we receive a physician-reported site of biopsy, based on which a case is classified as: local (breast biopsy), metastatic (other site biopsy) and lymph node. We have included a panel in Figure S6 (previously, Figure S5) and updated Table S9 (previously Table S7) to present the trend of PPA based on the site of biopsy for the tissue biopsy.

We also thank the reviewer for the suggestion to present the concordance results (PPA) based on receptor subtype. Figure S6 and Table S9 have been updated to include these analyses. The updated results are presented on page 5 of the revised manuscript.

- Figure 3A, the authors report multiple ctDNA-only variants that were not called in the tumor. How many of these variants had supporting reads in the matched tumor tissue but were not called as they were below the calling thresholds for the tissue assay?

We inspected the mapped bam files for the matched tumor tissue to see if there was sub-threshold evidence for the single base substitutions detected only in the matched liquid biopsy sample using *pysam* pileup. We inspected a total of 381 variants, of which 363 (>95%) had \leq five supporting reads, within the margin of sequencing error. These findings further support that a majority of these alterations are truly acquired, likely resistance, mutations detected in the liquid biopsy sample.

- Line 152: The authors report that “germline alterations were detected across a wide range of tumor fractions, as expected since the former can be detected in non-tumor DNA even when ctDNA shed is low)”. This is an obvious statement as the germline mutation calling is indeed expected to be more accurate when ctDNA fraction is low. The challenge for germline calling using the ctDNA-only approach is when the tumor fraction is high. The authors should report the accuracy of germline calling in high fraction ctDNA samples.

We thank the authors for these comments. The analysis was performed using a consensus germline approach whereby recurrent alterations that are nearly always flagged as germline in our dataset were labeled as ‘germline’. This was done to avoid the issue the authors mentioned of confounding issues with accuracy of germline calling. Our somatic/germline calling algorithm (PMID 29415044) has not yet been optimized/validated for use in liquid biopsy.

To avoid any confusion for readers, we have clarified this in figure legends for Figure S7 and S8 as well.

With regards to our statement, “germline alterations were detected across a wide range of tumor fractions, as expected since the former can be detected in non-tumor DNA even when ctDNA shed is low”, the key point of this statement was to indicate that all patients, regardless of shed rate could have these germline alterations detected (since they typically sit at a VAF of ~50%). To make this clearer, we have modified the statement to read: “Of note, germline alterations were detected, regardless of tumor fraction, as expected since the former can be detected in non-tumor DNA even when ctDNA shed is low (Figure S6D, Figure S7).”

- Figure S5C: Seems like this plot includes both germline or somatic BRCA1/2 mutations. I would suggest presenting data separately for germline and somatic variant in the same plot.

We now add a panel to the figure to present these results separately for *BRCA1/2* based on their predicted status (Figure S6E in the revised manuscript).

- Line 175-181: *The authors report identifying acquired amplification and rearrangements. Detection of large structural variants in ctDNA can often be challenging. Were these variants identified mainly in samples with high ctDNA fraction?*

Consistent with the reviewer’s comment, liquid biopsy samples with acquired amplifications and rearrangements had a significantly higher tumor fraction compared to those where these large, acquired events were not detected (28.6%, 26.5% vs. 2.0% respectively). To provide additional granularity on the ctDNA fraction in samples with acquired amplifications and rearrangements, we have presented the distribution of tumor fraction for each of these groups in Figure S9. The following text has been included on page 5 of the revised manuscript file –

“Of note, liquid biopsy samples with acquired amplifications and rearrangements had a significantly higher tumor fraction compared to samples where these large, acquired events were not detected (28.6%, 26.5% vs. 2.0% respectively, each $p < 10^{-5}$), thereby highlighting the potential of high tumor fraction LBx samples in detecting these complex events”

- Line 183: *The authors report multiple acquired PIK3CA mutations in ctDNA. The mutations appear to be the ones associated with APOBEC mutagenesis. What was the mutational burden for these ctDNA samples? It would be informative to perform mutational signature analysis on the tumor and/or ctDNA samples of these cases to assess for APOBEC associated mutational signatures 2 and 13.*

We thank the reviewer for the suggestion to explore mutational signatures, however our LBx assay capturing 62-70 genes poses a challenge for estimating blood-based TMB and signatures effectively. Therefore, to address this question, we instead explored the trinucleotide context for all acquired *PIK3CA* mutations in follow-up LBx samples in our paired cohort. These are presented in Table S12. Consistent with APOBEC-associated mutagenesis, a vast majority (48/62, 77%) of the single base substitutions were identified to be in APOBEC context. These results have also been added to Table S12 and page 6 of the revised manuscript.

- Figure 3D and lines 204-218:

1) It is important to highlight the performance of ctDNA in detecting the BRCA1/2 reversion variants as compared to the tumor and explicitly mention what fraction of these variants were also observed in the tumor or not.

2) What were the rates and patterns of BRCA1/2 reversion mutations in the full ctDNA cohort (3,339 samples)?

In the paired cohort, among patients with *BRCA1/2* reversions (Figure 3), none of the reversion alterations were observed in the tumor tissue biopsy specimen. This may be attributed to intervening treatments between the collection of the biopsies.

We also thank the reviewer for the suggestion to highlight the rates and patterns of *BRCA1/2* reversions in the full cohort of 3,339 LBx samples. To do so, we examined patients with two or more alterations in *BRCA1/2*. Among 10 patients with ≥ 2 *BRCA1* mutations, 8 showed mechanisms of reversions, and among 23 patients with ≥ 2 *BRCA2* mutations, 16 showed mechanisms of reversions. We have included this on pages 6-7 and Table S13 of the revised manuscript. The patients with 2 alterations and without evidence of reversion are cases with two pathogenic alterations (likely one hit on each *BRCA1/2* allele rather than a mutation with loss of heterozygosity).

- Line 220: The authors report that “Consistent with tumor heterogeneity and clonal evolution dynamics, acquired short variants in LBx revealed a lower clonal fraction relative to shared alterations with TBx”. The methods described in Tukachinsky et al and this manuscript, do not consider the copy number state of the variant and may over or underestimate the clonality of the ctDNA-only mutations. Would it be feasible to include the tumor-derived allele specific copy numbers in this analysis?

Unfortunately, quantitative copy number modeling in ctDNA samples is unreliable due to low rates of shed. When analyzing tissue samples, we indeed take this approach. Because of this limitation, we made the decision to show the entire distribution of estimated clonal fractions in Figure 3G. The differences are striking (median ~ 0.1 for acquired alterations and ~ 0.75 for shared alterations), and would not be explained by possible noise in the clonality measure.

- Figure 3F: This is an important finding and highlights the rate of polyclonal resistance in breast cancer. What are the patterns of multiple mutations for these alterations in the full ctDNA cohort stratified by receptor subtype (n=3,339)?

We observed similar patterns of polyclonal resistance in full ctDNA cohort, now shown in Supplementary Figure S11. While alterations in *ESR1*, *TP53*, *PIK3CA* and *NF1* were frequently polyclonal ($\sim 20\%+$), other gene alterations, such as in *AKT1*, *KRAS*, *EGFR* were predominantly monoclonal. Due to unavailability of receptor subtype information in our full ctDNA cohort, we were unable to perform this assessment based on receptor status. However, we extended the data presented in 3F to examine these patterns based on receptor subtype in the paired cohort (now presented in Figure S11). We observed polyclonal *TP53* mutations acquired across all subtypes whereas acquired polyclonal resistance mutations in *ESR1* were observed primarily in ER+ breast cancers.

- Lines 238-259: As the authors indicated, CH can potentially be a major source of somatic mutation in cfDNA. In addition to TP53 and JAK2 that are mentioned here, CH also frequently affects ATM, NF1, CHEK2 in cancer patients and can be detected in

cfDNA even in young individuals. These alterations are reported to be enriched in ctDNA (Fig 2B) and are often reported to be seen only in ctDNA (Fig 3A). How confident are the authors that these alterations are indeed tumor-derived and are not arising from CH. Are matched buffy coats available for assessment of CH for these patients? Additionally, non-recurrent CH mutations may affect non-canonical CH genes (Chabon et al, Nature 2020). BRCA1 and BRCA2 are large genes and some of their ctDNA-only non-pathogenic mutations could be passenger CH mutations. This issue needs to be further explored and discussed.

We agree that consideration of CH is important. We have modified the text of our CH section to include a discussion of *ATM*, *NF1*, *CHEK2*, *TP53*, and *JAK2*. There is strong evidence that *ATM*, *CHEK2*, and *JAK2* are frequently of CH origin based on liquid frequency, allelic fraction, and age differences. However, for *TP53* we are not observing these striking differences; when compared to metastatic tissue biopsies, alteration prevalences are similar. While *NF1* alterations were higher in prevalence in LBx compared to TBx, *NF1* is also a known resistance mechanism to endocrine therapy. We identified acquired *NF1* alterations in our longitudinal tissue biopsy cohort (Figure S12) as well as our CGDB (clinical) cohort, in samples taken post-endocrine therapy (Figure S13). This is covered on page 7 of the revised manuscript. We agree that matched buffy coats would be the best way to discern CH v non-CH, however these are not available. With regards to non-recurrent CH, since our analysis was limited to known or likely pathogenic alterations, incidental passenger alterations based on gene size should not be of concern. Specifically with regards to genes like *BRCA1/2* we observe nearly identical rates of alterations in tissue and liquid biopsies (Figure 1). We have also noted limitations of our study, with respect to CH, on pages 11-12 of the revised manuscript.

- CGDB analysis: It is not clear why the analysis of the CGDB cohort was primarily limited to only the patients with multiple tests, excluding >97% of the patients. The current analysis based on small numbers (196 patients: 104 with paired tumor tissues and 84 with paired tissue and ctDNA) provides some anecdotal evidence for enrichment of known mechanisms of resistance to certain therapies (MAPK alterations and ESR1 mut in 33 patients post-endocrine therapy, RB1 in 52 patients post-CDK4/6i and ERBB2 mut and PIK3CA mut in 23 patients after anti-HER2 therapy). The clinico-genomic analysis of the full cohort (1,150 patients with ctDNA data after excluding the cases with low purity and the 6,757 patients with tissue data) could have potentially provided robust associations with prior exposure to different classes of targeted therapies (endocrine therapy with AIs vs SERD vs SERm, CDK4/6i, PI3Ki, anti-HER2 Abs and TKIs, IO, etc) and shed light on potential novel mechanisms of resistance. The current analysis of the paired samples can be reported as a subset analysis of the full cohort to provide a rationale for serial ctDNA monitoring.

Analysis of paired samples in the CGDB allowed us to definitively identify which alterations were acquired (eg not present at baseline before a therapy and then emerged after treatment). However, we agree with the reviewer regarding the value of examining the full CGDB cohort. We have provided analyses that examine alteration prevalence with or without prior exposure to these agents (endocrine therapy, CDK4/6i therapy, PARPi, HER2-targeting therapy) and have added these analyses as Figure S13 and Table S17. This analysis identified associations of *ESR1* and *RB1* alterations in patients receiving a combination of endocrine therapy and CDK4/6i therapy, and *BRCA1/2* reversions with PARPi therapy, among other associations. We were limited for statistical analysis in some of these subgroups; a number of additional, rare gene alterations were elevated in post-treatment samples of specific therapy classes (Table

S17, Figure S13).

Case #1: ATM VAF is highly discordant with the rest of the alterations identified in the ctDNA and raises the possibility that this was potentially a CH variant. How does the authors explain this discordance?

We agree with the reviewer that this alteration is possibly of CH origin. We have modified the text to raise this possibility. We now write: "Mutations in *ATM* are common in solid tumors but are also in the bone marrow niche, raising the possibility of CH origin, although it is uncommon to detect CH variants above 10% of the cell-free DNA content"

Minor comments

- The number of samples and patients included in each analysis are not consistently reported in the legends and the text.

We have carefully reviewed the legends and the text to ensure this is reported accurately.

- The bar plots have the X-axis labels on top of the plot. It would be easier to follow the figure if these labels are placed below the X-axis line.

Bar plots in Figure 2, S2 and S5A have been updated to place the gene labels below the x-axis.

- Figure S2: It would be informative to add the percentages for TBx and LBx to the heatmap. The heatmap is also hard to follow. The colors are very close, and a large number of variants are coded as "multiple variants". I would suggest clarifying the colors or just use a standard oncoprint to presents the genomic landscape.

We have modified the colors on figure S2 to be more unique. In addition, we have added percentages to the right of the oncoplots.

- Line 105- 5% seems to be a typo and it should be 0.5%.

Thanks for identifying this typo. This has been fixed in the revised manuscript.

- Line 113: The short variant are defined in the methods. Please also define it in the text.

We have added this definition (point mutations and short indels) to the results section as well.

- Line 186: Would mention "qualified for PI3K inhibitors" rather than "alpelisib-qualifying context".

We have made this change.

- Line 211: BRCA should be BRCA2

We have ensured that all instances of "BRCA" were clarified as *BRCA1*, *BRCA2*, or *BRCA1/2*.

REVIEWER COMMENTS

Reviewer #1 (Remarks to the Author):

The authors have addressed all concerns of this reviewer. The manuscript is strengthened with the rewriting, rephrasing of some parts (in particular concerning "evolution"), the more precise descriptions of the cohorts and the statistics.

Reviewer #2 (Remarks to the Author):

Overall, the manuscript has markedly improved. The authors addressed most of my comments adequately and have modified the manuscript accordingly.

There are a few additional comments to be considered.

There are no legends provided for the main figures. The legends for the supplemental figures are often incomplete and do not include some of the information required to interpret the figures. For example, number of patients included in each analysis is not consistently reported. Some of the abbreviations and their explanations also appear to be missing or to be inaccurate (example Fig S13).

- The authors report: "For the above-described analysis, if multiple specimens were extracted from patients for the purpose of CGP testing and or patients had received multiple lines of therapy-in-question, the longest specimen collection date-therapy start date combination was chosen per patient". Why the longest interval between sample collection and start date of therapy was selected. Biologically it would have made sense to collect the pre-tx samples that were collected the closest to start date as pre-tx and the post-tx samples that were the closest to the end-date of therapy as post-tx.

- In the cohort description in the methods the authors report "Amongst 788 patients identified to be HER2 positive at metastatic diagnosis, 336 received a combination of hormone therapy and chemotherapy in the metastatic non-maintenance setting". Do they mean anti-HER2 therapy by "hormone therapy"? Does these analyses of HER2+ patterns really included hormone therapy

combinations instead of anti-HER2 therapy? This is a major in reporting or analysis, questioning the accuracies of the data and the analyses presented.

- Statistical methods is short and does not describe the performed analyses sufficiently. Are there any thresholds imposed to include genes/alterations in the comparative analyses? Are the P values one-sided or two-sided? What was the alpha threshold used to determine statistical significance? Are there any other statistical methods used beside Fisher's exact test? For example, how were difference between clonal fractions or tumor fractions analyzed?

- In the methods for comparison of paired TBx and LBx data, was the analysis limited to the intersect of the genomic regions (BED files) for both assays?

- Table S6 and Figure S3: The mutual exclusivity analysis has a major flaw and includes duplicate entries where G1 and G2 combinations are identical (e.g. there are entries for both ESR1-PIK3CA and PIK3CA and ESR1). The analysis also includes 18 rows where G1 and G2 are the same genes. These identical and duplicate rows need to be removed and the q values should be recalculated. There are also established algorithms such as CoMEt (Leiserson et al 2015) that could have been utilized to perform mutual exclusivity/cooccurrence analyses.

- it would be informative from the biological and clinical standpoints to include the subanalyses by receptor subtype in Fig S3.

- Lines 104-107: The authors report "Consistent with these differences, LBx showed higher similarity with metastatic-biopsied TBx compared to breast-biopsied TBx in the overall cohort, especially for genes like ESR1, NF1, and RB1, suggesting that some of these differences may be attributed to disease stage (Figure S5; Table S3)." Attributing these findings to stage alone seems to be an oversimplification as there were also significant difference in the number of previous lines of therapies, etc. The authors could have mentioned that these differences may be attributed to more advanced disease state at the time of testing.

- In response to my previous comments the authors have interrogated the tumor BAMs and report that "We inspected a total of 381 variants, of which 363 (>95%) had \leq five supporting reads, within the margin of sequencing error". This is worth being mentioned in the text and provided as a supplemental table. The method should also be described in the methods section.

- Figure S6F: There has been much interest in the site of metastatic disease and whether it can affect tumor ctDNA shedding. This large cohort of paired samples can potentially provide some exploratory

insight into this question. Focusing on the metastatic biopsies, was there a difference in PPAs for liver mets vs bone mets vs other sites of disease? Does the LN category include only the breast regional LNs or any metastatic LN (e.g., intra-abdominal or mediastinal LNs)?

- Table S11. There are large number of patients who are categorized in the “multiple” alterations category. This will limit the ability of the readers to better assess the rate of acquired mutations for each gene. The authors could also provide this information for each individual gene (similar to Table S10) by receptor subtype.

- Lines 171-176: The authors need to provide citations for each of these statements. It might also worth mentioning that the ERBB2 mutations may also predict response to HER2-ADCs (Li et al, Cancer Discovery 2020 and NEJM 2022)

- The authors report "There was a trend towards higher reversion rates in BRCA1 relative to BRCA2 (25% v 19%, $p=0.7$)". This is somehow misleading given the number of events and the 95% CI around these frequencies.

- Comparing Figure S12 and 3C, there appears to be less tissue samples in the “multiple” alterations category than in liquid biopsy samples. Is this difference statistically significant? This could again highlight the utility of LBx in detection of polyclonal events involving different genes.

- Fig S13 and Table S17: Some of the abbreviations and their explanations are incoherent. What are “HT” and “HER2 Hormone Therapy”?

This analysis needs to be revised and reviewed. Suggest using the treatment paradigms used in Figure 4B.

What is “R” next BRCA1 and BRCA2? Is this referring to a reversion mutation?

Suggest using CDK4/6i instead of CDK in both Fig 4B and the supplemental tables.

- Table S17:

a) The ORs are calculated comparing pre with post. It would be more logical if they compared post vs pre. Please also provide the 95% CI s for the ORs.

b) What was the rationale for not adjusting the P values for multiple testing in this analysis?

c) Was there a frequency threshold (or number of altered genes in the cohort) above which the genes were included in this analysis. Seems like the analysis also includes alterations with very small

frequencies. These comparisons are obviously underpowered and excluding them would have improved the statistical power of such analysis.

- Case #2: Regarding the homozygous BRCA1 variant, did the authors mean that the somatic BRCA1 variant was accompanied by LOH of WT allele resulting in biallelic BRCA1 loss or that the mutation is truly homozygous (i.e. biallelic) that could potentially happen in the setting of WGD, etc. This needs to be clarified.

- Case #3: It is not clear what the "liver metastasis" means in the case description. Does it mean that the patient had progression in the liver following gem-HP?

Please note that acquired ERBB2 mutations may be mechanisms of resistance to HER2-TKIs in patients with harboring ERBB2 mutations at baseline (Smyth et al, Cancer discovery 2020). MAPK alterations (including ERBB2 and KRAS mutations) can also confer resistance to anti-HER2 therapy among the patients with HER2 amplified tumors (Smith et al, Nat Com 2021)

Minor comments

- The order of supplemental figures and tables does not match the order they were presented in the text.

- Line 87: The frequencies for ERBB2 seems to be not in order. It should be (3.4%, 4.3%) if they follow the same patterns as the other genes (TBx, LBx)

- Lines 95-97: The authors report "In an analysis limited to the most recent assay targeting 324 genes (n = 1,430), similar results were seen on this platform, particularly for samples with tumor fraction of \geq 10%. (Figure S4, Table S7)." This sentence seems to be misplaced and has no connect with the immediately preceding section covering mutual exclusivity analysis.

- The supplemental tables often only include the percentages and do not include the exact counts.

- Fig S1: It is not clear why BRCA1/2 have been labeled in all the plots. It would have been more informative to label the genes that are biologically relevant but do not reach statistical significance in this subanalysis (such as RB1, PTEN, NF1, etc).

- Line 336. Please note that BRAF V600E has now pan-cancer FDA accelerated approval for dabrafenib/trametinib combination therapy.

RESPONSE TO REVIEWERS

Reviewer #1 (Remarks to the Author):

The authors have addressed all concerns of this reviewer. The manuscript is strengthened with the rewriting, rephrasing of some parts (in particular concerning "evolution"), the more precise descriptions of the cohorts and the statistics.

We thank the reviewer for taking the time to review and for their thoughtful feedback to strengthen our manuscript.

Reviewer #2 (Remarks to the Author):

Overall, the manuscript has markedly improved. The authors addressed most of my comments adequately and have modified the manuscript accordingly.

There are a few additional comments to be considered.

We thank the reviewer for their detailed review and feedback. We have addressed the additional comments below.

There are no legends provided for the main figures. The legends for the supplemental figures are often incomplete and do not include some of the information required to interpret the figures. For example, number of patients included in each analysis is not consistently reported. Some of the abbreviations and their explanations also appear to be missing or to be inaccurate (example Fig S13).

We uploaded the legends during submission and apologize for it not making it into the merged manuscript files. We have rectified this in the resubmission. We have also provided more details in all the supplemental figures and addressed missing information in Figure S14 (previously Figure S13).

- The authors report: "For the above-described analysis, if multiple specimens were extracted from patients for the purpose of CGP testing and or patients had received multiple lines of therapy-in-question, the longest specimen collection date-therapy start date combination was chosen per patient". Why the longest interval between sample collection and start date of therapy was selected. Biologically it would have made sense to collect the pre-tx samples that were collected the closest to start date as pre-tx and the post-tx samples that were the closest to the end-date of therapy as post-tx.

We thank the reviewer for their comment. First, we would like to highlight the rarity of having to choose among multiple samples. Due to insurance coverage, how physicians practice medicine, and other considerations, most patients with multiple tests had precisely two tests for examination (204 patients), with only 27 patients requiring a choice of sample. We chose the longest time interval to be consistent with the analyses performed in the overall dataset (Figures 1-3). The longest time interval allows for emerging resistance mutations to become fixed in the population as clonal.

- In the cohort description in the methods the authors report "Amongst 788 patients identified to be HER2 positive at metastatic diagnosis, 336 received a combination of hormone therapy and chemotherapy in the metastatic non-maintenance setting". Do they

mean anti-HER2 therapy by “hormone therapy”? Does these analyses of HER2+ patterns really included hormone therapy combinations instead of anti-HER2 therapy? This is a major in reporting or analysis, questioning the accuracies of the data and the analyses presented.

We inadvertently wrote hormone therapy rather than HER2-targeting therapy (HT v HT) and apologize for the lack of clarity here. As the reviewer noted, these patients received HER2-targeted therapy. To further enhance this section, we examined the patterns of alterations split by HER2 tyrosine kinase inhibitors and antibody drug conjugates. We have updated the methods section to say the following: “Amongst 788 patients identified to be HER2 positive at metastatic diagnosis, 304 received a combination of HER2 antibody (Ab)-based targeted therapy and chemotherapy in the metastatic non-maintenance setting (192 patients had pre-treatment biopsies and 144 had post-treatment biopsies) and 107 received a combination of HER2 TKI and chemotherapy in the metastatic non-maintenance setting (73 patients had pre-treatment biopsies and 39 had post-treatment biopsies).”.

- Statistical methods is short and does not describe the performed analyses sufficiently. Are there any thresholds imposed to include genes/alterations in the comparative analyses? Are the P values one-sided or two-sided? What was the alpha threshold used to determine statistical significance? Are there any other statistical methods used beside Fisher’s exact test? For example, how were difference between clonal fractions or tumor fractions analyzed?

The statistical methods section has been expanded to include the details requested by the reviewer.

“Differences in prevalence of gene alterations between TBx and LBx assays, as well as patterns of co-occurrence and mutual exclusivity between gene alterations were tested using a Fisher’s exact test. Two-sided P-values were calculated for each comparison and then adjusted for multiple hypothesis testing using the Benjamini-Hochberg FDR method. For continuous variables (e.g., clonal fraction, LBx tumor fraction), Wilcoxon rank sum test was used to test for differences between specific groups; two-sided P-values were calculated for each comparison. For all analyses, the significance level was set to 0.05. Statistics, computation, and plotting were carried out using Python 2.7 (Python Software Foundation) and R 3.6.1 (R Foundation for Statistical Computing).”

In general, most analyses included the 70 genes covered in the LBx assay (Table S2). Additional criteria (if any) applied for specific analysis have been reported alongside the corresponding supplementary tables.

- In the methods for comparison of paired TBx and LBx data, was the analysis limited to the intersect of the genomic regions (BED files) for both assays?

The analysis was limited to the genomic regions that are covered in both assays. We further clarify this in the Methods section. We have included the following sentence:

“Further, comparisons between TBx and LBx were limited to the genomic regions covered in the TBx and LBx assays within these 70 genes.”

- Table S6 and Figure S3: The mutual exclusivity analysis has a major flaw and includes duplicate entries where G1 and G2 combinations are identical (e.g. there are entries for

both ESR1-PIK3CA and PIK3CA and ESR1). The analysis also includes 18 rows where G1 and G2 are the same genes. These identical and duplicate rows need to be removed and the q values should be recalculated. There are also established algorithms such as CoMEt (Leiserson et al 2015) that could have been utilized to perform mutual exclusivity/cooccurrence analyses.

We thank the reviewer for their detailed review of our supplemental materials. We can understand why the reviewer believes the analysis to be flawed, however, the “duplicate entries” are intentional to allow for easier analysis of the data by readers. If a reader wants to understand what is mutually exclusive for *ESR1*, then they can filter our data for gene1 = *ESR1* and get all co-occurrence analyses. Otherwise, datasets would have to be filtered and combined based on gene1 = *ESR1* and gene2 = *ESR1*. We have however excluded listing of the self-gene pairs. We confirm that the FDR q-values displayed are accurate based on the full list of co-occurrence analyses.

- it would be informative from the biological and clinical standpoints to include the subanalyses by receptor subtype in Fig S3.

We agree that analyses based on receptor subtype are valuable. We have therefore added a new supplemental figure (Figure S2) examining the overlap of gene alterations in each receptor subtype group.

- Lines 104-107: The authors report "Consistent with these differences, LBx showed higher similarity with metastatic-biopsied TBx compared to breast-biopsied TBx in the overall cohort, especially for genes like ESR1, NF1, and RB1, suggesting that some of these differences may be attributed to disease stage (Figure S5; Table S3)." Attributing these findings to stage alone seems to be an oversimplification as there were also significant difference in the number of previous lines of therapies, etc. The authors could have mentioned that these differences may be attributed to more advanced disease state at the time of testing.

We agree with the reviewer that our statement was an oversimplification. Disease stage is likely one major factor, but other factors like lines of therapy are likely key in explaining the prevalence differences between liquid and tissue (as highlighted by Table 1). We have edited this section to reflect the complexities, now writing:

“Consistent with these differences, LBx showed higher similarity with metastatic-biopsied TBx relative to breast-biopsied TBx in the overall cohort, especially for genes like *ESR1*, *NF1*, and *RB1* (Figure S6; Table S3). This suggests that some of these differences in alteration prevalence may be attributed to disease stage. However, since most resistance-associated alterations were still observed at a significantly higher prevalence in LBx compared to metastatic-biopsied TBx, other factors are likely influencing these findings, including the number of lines of prior therapy.”

(Please note the figure number change: Figure S5 is now Figure S6)

- In response to my previous comments the authors have interrogated the tumor BAMs and report that "We inspected a total of 381 variants, of which 363 (>95%) had ≤ five supporting reads, within the margin of sequencing error". This is worth being mentioned

in the text and provided as a supplemental table. The method should also be described in the methods section.

We have included this analysis in the Results and the Methods sections of the revised manuscript. We have also included a supplemental table summarizing these results (Table S14 of the revised manuscript).

- Figure S6F: There has been much interest in the site of metastatic disease and whether it can affect tumor ctDNA shedding. This large cohort of paired samples can potentially provide some exploratory insight into this question. Focusing on the metastatic biopsies, was there a difference in PPAs for liver mets vs bone mets vs other sites of disease? Does the LN category include only the breast regional LNs or any metastatic LN (e.g., intra-abdominal or mediastinal LNs)?

Understanding patterns in the site of metastatic disease, from the context of ctDNA shedding is an interesting area to investigate further. Although it is beyond the scope of this analysis, a preliminary analysis of metastatic tumor biopsies sites with at least 50 samples (liver and bone metastasis) revealed similar PPA - 86% in liver, 84% in bone biopsies. Table S9 has been updated to include these results.

Lymph node biopsies may include loco-regional or distant biopsies. The site of biopsy is typically reported as part of the test requisition form by the ordering physician. Therefore, we do not have full information on where the lymph node biopsies were procured.

- Table S11. There are large number of patients who are categorized in the “multiple” alterations category. This will limit the ability of the readers to better assess the rate of acquired mutations for each gene. The authors could also provide this information for each individual gene (similar to Table S10) by receptor subtype.

While it is valuable to see the large number of patients categorized in the “multiple” group, we agree that it limits visibility into the rate of acquired mutations for each gene. We have therefore added a detailed summary table as part of Table S11, with the number of cases exhibiting an acquired alteration in each gene assessed for both analyses – (i) based on receptor subtype, and (ii) based on time between the tests. To maintain consistency, Table S16 was also updated to match the revised Table S11.

- Lines 171-176: The authors need to provide citations for each of these statements. It might also worth mentioning that the ERBB2 mutations may also predict response to HER2-ADCs (Li et al, Cancer Discovery 2020 and NEJM 2022)

We have updated the text to include references and a mention of the possible use of HER2-ADCs in this population.

- The authors report "There was a trend towards higher reversion rates in BRCA1 relative to BRCA2 (25% v 19%, p=0.7)". This is somehow misleading given the number of events and the 95% CI around these frequencies.

We agree that this is misleading given the wide confidence intervals and high p-value (0.7). We have modified the text to state that the reversion rates were similar, now writing:

“There were similar reversion rates in *BRCA1* and *BRCA2* (25% v 19%, p=0.7).”

- Comparing Figure S12 and 3C, there appears to be less tissue samples in the “multiple” alterations category than in liquid biopsy samples. Is this difference statistically significant? This could again highlight the utility of LBx in detection of polyclonal events involving different genes.

We thank the reviewer for their keen eye. Since liquid biopsies integrate the DNA from multiple resistance sites/clones this platform has the potential to identify polyclonal resistance at a higher rate than tissue biopsies which are taken from a single biopsy site. In figure 3H we demonstrate high rates of polyclonal alterations *within* a gene, but as the reviewer points out, comparing Figure S13 (previously Figure S12) to Figure 3C suggests high rates of polyclonal events involving different genes. We quantified these differences using a Fisher’s exact test and added a statement in the manuscript regarding this finding:

“Interestingly, a higher rate of acquired events in multiple genes was observed in follow-up LBx compared to TBx (150 follow-up LBx samples, 21% vs. 98 follow-up TBx samples, 7.3%; $p < 10^{-5}$, Table S11, Table S16).”

- Fig S13 and Table S17: Some of the abbreviations and their explanations are incoherent. What are “HT” and “HER2 Hormone Therapy”? This analysis needs to be revised and reviewed. Suggest using the treatment paradigms used in Figure 4B. What is “R” next BRCA1 and BRCA2? Is this referring to a reversion mutation? Suggest using CDK4/6i instead of CDK in both Fig 4B and the supplemental tables.

We thank the reviewer for their attention to detail in this analysis. We have expanded the abbreviations and provided additional explanation to ensure accuracy of the treatment paradigms. Additionally, we assessed the HER targeted therapy group in two sub-categories – antibody-drug conjugates and tyrosine kinase inhibitors to keep it consistent with Figure 4B; unfortunately, we were limited by cohort size and underpowered for statistical analyses within these subcategories.

BRCA1^R and *BRCA2^R* indicate reversion alterations only, for the PARPi group. We have included this explanation in the revised Figure S14 (previously Figure S13) and Table S18 (previously Table S17), and their associated legends. We have also used CDK4/6i instead of CDK in Figure 4B, Figure S14, Table S17 and Table S18.

- Table S17:

- a) The ORs are calculated comparing pre with post. It would be more logical if they compared post vs pre. Please also provide the 95% CI s for the ORs.**
- b) What was the rationale for not adjusting the P values for multiple testing in this analysis?**
- c) Was there a frequency threshold (or number of altered genes in the cohort) above which the genes were included in this analysis. Seems like the analysis also includes alterations with very small frequencies. These comparisons are obviously underpowered**

and excluding them would have improved the statistical power of such analysis.

We thank the reviewer for their feedback on this analysis.

- a) We agree and have calculated the Odds ratio (OR) as suggested. We also include the 95% confidence intervals for the OR.
- b) Given the exploratory nature of this analysis and the limited cohort size, we had previously provided raw P-values. We have however also included the FDR adjusted P-values in the revised table.
- c) In the interest of sharing all identified gene alterations, we had not applied any frequency threshold. We continue to share all the identified gene alterations; however, the statistical analysis was only performed in cases where we observed a recurrent gene alteration, defined as being observed in at least 2 cases.

The updated analysis can be found in Table S18 (previously Table S17).

- Case #2: Regarding the homozygous BRCA1 variant, did the authors mean that the somatic BRCA1 variant was accompanied by LOH of WT allele resulting in biallelic BRCA1 loss or that the mutation is truly homozygous (i.e. biallelic) that could potentially happen in the setting of WGD, etc. This needs to be clarified.

The text has been changed to “Tissue CGP revealed a L1620*fs in exon 10 of *BRCA1* (predicted somatic, with loss of heterozygosity of the wild type allele),” for clarity.

The prediction was done using a validated somatic-germline-zygosity prediction algorithm (reference: PMID: 29415044). In brief, the algorithm predicts whether a variant is germline or somatic, and how many mutant versus wild type alleles are present in the tumor by modeling the alteration's allele frequency, taking into account the tumor content, tumor ploidy, and the local copy number. In this particular case, LOH appears to be a result of a copy number loss of the WT allele (not copy-neutral LOH).

- Case #3: It is not clear what the “liver metastasis” means in the case description. Does it mean that the patient had progression in the liver following gem-HP? Please note that acquired ERBB2 mutations may be mechanisms of resistance to HER2-TKIs in patients with harboring ERBB2 mutations at baseline (Smyth et al, Cancer discovery 2020). MAPK alterations (including ERBB2 and KRAS mutations) can also confer resistance to anti-HER2 therapy among the patients with HER2 amplified tumors (Smith et al, Nat Com 2021)

We thank the reviewer for pointing out the confusing wording. The text has been changed to “The patient was switched to paclitaxel/pertuzumab/trastuzumab, and then gemcitabine/pertuzumab/trastuzumab. A TBx (liver metastasis) and a LBx were collected and profiled during this last treatment.”

We have added the references to additional evidence about *de novo* and acquired resistance to anti-HER2 therapy.

Minor comments

- The order of supplemental figures and tables does not match the order they were presented in the text.

We have confirmed that all supplemental figures and tables are listed in order of occurrence in the text.

- Line 87: The frequencies for ERBB2 seems to be not in order. It should be (3.4%, 4.3%) if they follow the same patterns as the other genes (TBx, LBx)

Thank you for catching this error, we have fixed the percentages to (3.4%, 4.3%) to match the (TBx, LBx) order.

- Lines 95-97: The authors report "In an analysis limited to the most recent assay targeting 324 genes (n = 1,430), similar results were seen on this platform, particularly for samples with tumor fraction of $\geq 10\%$. (Figure S4, Table S7)." This sentence seems to be misplaced and has no connect with the immediately preceding section covering mutual exclusivity analysis.

We understand that this sentence, as written, may be misinterpreted as an extension of the mutual exclusivity analysis. Therefore, we have reworded it to be clearer.

"In an analysis of the most recent assay targeting 324 genes (n = 1,430)²², patterns of gene alteration prevalence were similar to those observed in the larger LBx cohort (n = 3,339), particularly for samples with tumor fraction of $\geq 10\%$. (Figure S5, Table S7)."

(Figure number change: Figure S4 to Figure S5)

- The supplemental tables often only include the percentages and do not include the exact counts.

We have included the total number of samples in each cohort as well as exact counts in addition to the percentages, where applicable, in the revised supplementary tables.

- Fig S1: It is not clear why BRCA1/2 have been labeled in all the plots. It would have been more informative to label the genes that are biologically relevant but do not reach statistical significance in this subanalysis (such as RB1, PTEN, NF1, etc).

We had provided a supplementary table (Table S4) with the prevalence of all gene alterations assessed, to accompany this figure. We had originally labeled genes with statistically significant difference in prevalence between the platforms as well as a few key driver gene alterations. We have now also labelled additional biologically relevant genes in the figure (e.g., *ESR1*, *RB1*, *NF1*, *PTEN*), to make this figure more informative.

- Line 336. Please note that BRAF V600E has now pan-cancer FDA accelerated approval for dabrafenib/trametinib combination therapy.

This was indeed a very exciting development since the original submission of our manuscript. We have added a sentence to highlight this:

“*BRAF* V600E was recurrently observed as an acquired alteration, notably since the FDA granted accelerated approval for dabrafenib/trametinib combination therapy.”

REVIEWERS' COMMENTS

Reviewer #2 (Remarks to the Author):

The authors have addressed the comments and concerns of this reviewer adequately. I have no further comments.